

Geographically divergent trends in snowmelt timing and fire ignitions across boreal
North America
Thomas D. Hessilt[1], Brendan M. Rogers[2], Rebecca C. Scholten[1], Stefano Potter[2], Thomas A.J. Janssen[1]
and Sander Veraverbeke[1]
1 Faculty of Science, Vrije Universiteit Amsterdam, Amsterdam, The Netherlands
2 Woodwell Climate Research Center, Falmouth, MA, USA
**Correspondence**: Thomas D. Hessilt (t.d.hessilt@vu.nl)



**Abstract**
The snow cover extent across the Northern Hemisphere has diminished while fire extent and severity has
increased over the last five decades with accelerated warming. However, the effects of earlier snowmelt
on fire is largely unknown. Here, we assessed the influence of snowmelt timing on fire ignitions across 16
ecoregions of boreal North America. We found spatially divergent trends in earlier (later) snowmelt led to
an increasing (decreasing) number of ignitions for the northwestern (southeastern) ecoregions between
1980 and 2019. Similar northwest-southeast divergent trends were observed in the changing length of the
snow-free season and correspondingly the fire season length. We observed increases (decreases) over
Northwest (Southeast) boreal North America which coincided with a continental dipole in air temperature
changes between 2001 and 2019. Earlier snowmelt induced earlier ignitions of between 0.22 and 1.43
days earlier per day of earlier snowmelt in all ecoregions between 2001 to 2019. Early-season ignitions
(defined by the 20 % earliest fires per year) developed into significantly larger fires in 8 out of 16
ecoregions and 77 % larger across the whole domain. Using a piecewise structural equation model, we
found that earlier snowmelt is a proxy for earlier ignitions but may also result in a cascade of effects from
earlier desiccation of fuels and favorable weather conditions that led to earlier ignitions. This indicates
that snowmelt timing is an important trigger of land-atmosphere dynamics. Future warming and
consequent changes in snowmelt timing may contribute to further increases in western boreal fires while
the number of fires in eastern boreal North America may increase too with climate change.
**1. Introduction**
Snow cover across boreal and Arctic ecosystems is an important driver of regional hydrological cycles
and the global energy balance (Swenson and Lawrence, 2012; Li et al., 2017). With climate warming,
spring snow cover has decreased 11 % per decade over the Northern Hemisphere since 1970s (Déry and
Brown, 2007; Brown and Robinson, 2011). Changes in snow cover and sea ice have led to a substantial
decrease in the cryosphere radiative forcing across the Northern Hemisphere of around 0.5 W m$^{-2}$ from
1979 to 2008 which warms regional and global climate (Flanner et al., 2011; Groisman, et al., 1994). This



feedback importantly contributes to accelerated warming in the northern high latitudes (Anisimov et al.,
2007; Rantanen et al., 2022). However, the changes in snow cover are heterogeneous across the Northern
Hemisphere. Over boreal North America, changes in snow cover timing show a long-term spatial
divergence between earlier (later) snowmelt timing over western (eastern) boreal North America between
1972 and 2017 (Chen et al., 2016; Bormann et al., 2018). The divergent changes in snow cover will likely
have important impacts on ecosystem functioning in boreal forest and Arctic tundra (Post et al., 2009;
Buermann et al., 2013) and may potentially be attributed to persistent changes in atmospheric circulations
(Jain and Flannigan, 2021).
Simultaneously, over the last two decades, large parts of western boreal North America have
experienced a rise in the number of lightning fire ignitions and burned area, driven by increases in dry
fuel availability (Abatzoglou et al., 2016; Hessilt et al., 2022), favorable fire weather (Sedano and
Randerson, 2014), and increase in the number of lightning strikes (Veraverbeke et al., 2017). Fire is the
most widespread ecosystem disturbance change in boreal North America and these increasing trends in
fire occurrence are expected to continue in the future (Flannigan et al., 2005; Balshi et al., 2009; Chen et
al., 2021; Phillips et al., 2022). Early snowmelt has previously been linked to large fires in the western
United States as a consequence of longer periods of fuel drying (Westerling et al., 2006). Dry fuel
availability is a prerequisite for fire ignitions (Abatzoglou et al., 2016; Hessilt et al., 2022), and may
further enable rapid fire growth thereby resulting in larger fires (Sedano and Randerson, 2014;
Veraverbeke et al., 2017). The relationships between snowmelt timing and fire behavior characteristics,
such as fire ignitions and size, may vary across boreal North America and remain poorly understood.
In recent years, early snowmelt after warm winters has been linked to summer heatwaves and
severe fire seasons over Siberia (Gloege et al., 2022; Scholten et al., 2022). Warm winter extremes can
substantially impact ecosystem functioning until deep into the subsequent growing season (Zona et al.,
2022). Early snowmelt induces an early vegetation green-up because of early peaks in soil moisture
(Gloege et al., 2022) but a decreased late season vegetation productivity (Buermann et al., 2013; Miles



and Esau, 2016; Graham et al., 2017). The enhanced evapotranspiration can lead to soil desiccation in
spring and result in increased sensible heat flux later in spring (Gloege et al., 2022). This enhances
atmospheric warming and drying through limited evaporative cooling (Seneviratne et al., 2010). In turn,
positive geopotential height anomalies and persistent atmospheric ridge formations (Cohen et al., 2014;
Tang et al., 2014; Jain and Flannigan, 2021) that promote atmospheric blocking events thereby create
favorable weather conditions for fire ignition and spread (Coumou et al., 2018; Jain and Flannigan, 2021;
Scholten et al., 2022). Simultaneously, destabilization of the atmosphere increases the occurrence of
convective thunderstorms and lightning (Chen et al., 2021), and the increases in cloud-to-ground
lightning strikes potentially increasing the likelihood of igniting dry fuels (Hessilt et al., 2022).
Nonetheless, the influence of a divergent snow cover trend across boreal North America on weather, fuel
dryness, and ignition timing has previously not been studied and may exhibit divergent responses to
changes in the snow cover.

Earlier snowmelt may also lead to an earlier start of the fire season and possibly more severe fire

weather, thereby lengthening and intensifying the boreal fire season (Flannigan et al., 2005; Veraverbeke
et al., 2017). Different methods have been used to quantify the length of the boreal fire season. The fire
season length has been estimated using fire weather indices as proxies of fire activity (Wotton and
Flannigan, 1993; Flannigan et al., 2016). Other studies have estimated the fire season length using long-
term government records, which are prone to temporal changes in accuracy and uncertainties (Hanes et
al., 2019). Daily fire monitoring using the polar-orbiting Moderate Resolution Imaging Spectroradiometer
(MODIS) sensors allows accurate definition of the fire season based on observed fire activity since the
2000s (Justice et al., 2002; Giglio et al., 2016, 2018). Given that the MODIS record dates back until the
early 2000s, it may be possible to infer changes in fire season length across boreal North America during
this period.

Here, we investigated relationships between snowmelt and early season ignition timing across

boreal North America between 2001 and 2019. In addition, we evaluated the influence of ignition timing





on fire size and assessed temporal changes in snowmelt timing and the number of ignitions since 1980.
Through satellite-derived estimates, we derived the length of the snow-free and the fire seasons, and
assessed the influence of the length of the snow-free season on fire season length. Early ignition timing
was modeled as a function of snowmelt timing, and meteorological and fire weather conditions using a
linear mixed-effect model to investigate potential cascading effect of earlier snowmelt timing. Finally, we
assessed the interactions between snowmelt timing, and meteorological and fire weather conditions when
modeling ignition timing through a piecewise structural equation model.

## 2. Methodology

*2.1 Study domain*

The study domain includes Alaska, USA, and the majority of Canada ($9.17 \times 10^6$ km$^2$) excluding the
Canadian Arctic Archipelago, and is divided into sixteen ecoregions (Omernik, 1987, 1995) (Fig. 1). We
used the second-level ecoregions for subcontinental comparisons (McCoy and Neumark-Gaudet, 2022).
We included 14 ecoregions but further divided the Softwood Shield and Taiga Shield into eastern and
western ecoregions due to their large longitudinal gradients, resulting in 16 different ecoregions in our
study (Fig. 1 and Table S1). The Softwood Shield was divided in accordance with the third-level
ecoregion division and the Taiga Shield was split into two sub-regions East and West of Hudson Bay
(Baltzer et al., 2021) (Fig. 1). The northernmost ecoregions (the Arctic Cordillera, Northern Arctic, and
Southern Arctic) were excluded as they included very few ignitions. The southern parts of the Cold
Deserts, Marine West Coast Forest, Mixed Wood Shield, and Western Cordillera were cropped out as
they were not covered by the Arctic-Boreal Vulnerability Experiment Fire Emission Database (ABoVE-
FED; Potter et al., 2023) extent (Fig. 1). Our study domain thus included Arctic tundra and boreal forest
ecosystems between Northwest Alaska and Southeast Canada.



*2.2 Snowmelt timing*
We retrieved snowmelt timing at 463 m resolution from the MODIS daily composite snow-cover product
MOD10A1 collection 6 between 2001 to 2019 (Hall and Riggs, 2016). This product computes the
normalized difference snow index (NDSI) ranging from 0 to 1 from visible and shortwave infrared
spectral data. The relationship between NDSI and estimated fractional snow cover from higher resolution
snow cover data from Landsat Enhanced Thematic Mapper-plus (30 m) has previously been proven
robust over large areas such as boreal North America (Salomonson and Appel, 2004). This allowed us to
use NDSI as a proxy for fractional snow cover. We identified the Julian calendar day of snowmelt timing
as the first day a pixel had less than or equal to 15 % snow cover for a minimum of 14 consecutive days
(Verbyla, 2017). Pixels that contained persistent cloud cover, water, or perennial snow cover (more than
250 days a year), or less or equal than 15 % snow cover for less than 14 consecutive days were excluded
from the analysis. Pixels with values exceeding a pixel-specific threshold (average snowmelt timing in
2001-2019 ± 3 standard deviations) were regarded as outliers and excluded from the analysis. The
snowmelt timing was determined between February 1 and July 31. We opted for a large potential range in
snowmelt timing because of the large latitudinal and thus climatological range present in the study
domain (Fig. 1). To retrieve the first day of snow cover, we used the reversed method where the first day
on which at least 15 % of the pixel was snow covered for a minimum of 14 consecutive days was set to
first day of snow cover. This was determined between August 1 and December 31. We modified the code
from Armstrong et al. (2023) to compute the snowmelt timing in Google Earth Engine.

In complement to the MODIS snow cover product, we also used the Northern Hemisphere Equal-

Area Scalable Earth Grid 2.0 version 4 weekly snow cover product (NSIDC) to calculate long-term
snowmelt timing and snow cover onset trends since 1980 (Brodzik and Armstrong, 2013; Estilow et al.,
2015). The NSIDC product is based on the National Oceanic and Atmospheric Administration (NOAA)
climate data record (Robinson et al., 2012). It uses visual interpretation of snow cover detected from a
range of sensors (i.e. Advanced Very High Resolution Radiometer (AVHRR), Geostationary Operational
Environmental Satellite (GOES), and more recently MODIS (Helfrich et al., 2007)) and interpolated to



the Equal-Area Scalable Earth (EASE) grid of 25 km. The NSIDC product is influenced by image
availability and user interpretation of images (Ramsay, 1998; Helfrich et al., 2007). It uses a binary
indication of snow or no snow cover. We therefore computed the annual first day with no snow cover for
all pixels. Similar to the MODIS product, the snowmelt timing was determined between February 1 and
July 31. The MODIS and NSIDC snow cover products differ both in their temporal and spatial
resolutions, but we found reasonable agreement between snowmelt timing from both products across the
study domain (RMSE = 12.57 Julian day, r = 0.76 $p$ < 0.01) and individual ecoregions (Fig. S1).

*2.3 Fire information*
The location and timing of the fire ignitions, and their associated burned area, were derived from the
Arctic-Boreal Vulnerability Experiment Fire Emission Database (ABoVE-FED) product (Potter et al.,
2023). The ABoVE-FED burned area product covers Alaska and Canada (2001-2019) and is derived from
thresholding the differenced normalized burn ratio (dNBR) from Landsat imagery at 30 m resolution
complemented by MODIS surface reflectance products at 500 m resolution (MOD09GA and MYD09GY
v6) when no Landsat data were available. The dNBR thresholding within the ABoVE-FED product was
limited to the fire perimeters from the Alaskan Large Fire Database (ALFD, (Kasischke et al., 2002)) and
Canadian Large Fire Databases (CLFD, (Stocks et al., 2002)) and MODIS active fire locations, and their
surroundings, to minimize commission errors from non-fire disturbances (Veraverbeke et al., 2015; Potter
et al., 2023).

The retrieval of ignition timing and location was adapted from Scholten et al. (2021b). This

algorithm uses the spatiotemporal information in the ABoVE-FED burned area product to delineate
individual fire perimeters and a minimum search radius to detect the location of each unique ignition
spatially and temporally. Since burned area pixels in boreal regions can be discontinuous due to varying
fire severity and possibly omitted pixels, we applied different buffers (1 km and 2 km) to group the fire
pixels into fire perimeters. Several combinations of the fire perimeter buffers (1 km and 2 km), search



radii (5 km, 7.5 km, 10 km, and 15 km), and minimum fire sizes (i.e., exclusion of fires from 1 or 2
individual burned pixels) were examined to minimize the commission and omission errors. We tested
these three fire size thresholds, as single or double pixel burned area could be small anthropogenic fires or
commission errors. We compared the results to the ignitions present in the Alaskan Fire Emission
Database (AKFED) version 2 (Scholten et al., 2021a) (Table S2). We used ignition locations and timing
retrieved inside 2 km buffered fire perimeters, using a 7.5 km search radius for fires larger than 50 ha (1
and 2 pixel fires removed) as this was in good agreement with the AKFED-derived ignitions (Table S2).
This led to an exclusion of 15 % ignition locations compared to an inclusion of all fire sizes. In Alaska,
Yukon, and the Canadian Northwest Territories, we found approximately 6 % more ignitions in ABoVE-
FED compared to AKFED, and 76 % overlap between the two ignition datasets.
For this study, we also removed ignition locations that were not covered by snow between 2001
and 2019 and ignitions that were erroneously detected before snowmelt (approximately 11 % of the
observations). For the whole study domain and period, we analyzed a total of 17 957 ignitions (Fig. 1b).
When possible, we assigned the ignition cause, lightning or anthropogenic, from the ignition cause
attribute of the ALFD and CLFD when ignitions fell within the fire perimeter from the same year. By
doing so, 4 % of the ignitions were attributed an anthropogenic cause, 38 % were attributed a lightning
cause, and the cause of the remaining 58 % was unknown. The daily timing and exact location of fire
ignitions were derived from the ABoVE-FED data between 2001 and 2019, but we extended the number
of ignitions within ecoregions back to 1980 using fire perimeter data from the ALFD and CLFD. The start
year 1980 was chosen as it corresponds to major optimization of lightning detection systems for Canada
that minimized erroneous attribution of causes to fires (Stocks et al., 2002).
We established a relationship between the number of ignitions from ABoVE-FED and the
number of fire perimeters from the ALFD and CLFD for the overlapping period between 2001 and 2019
per ecoregion (Fig. S2). The statistical relationship between the number of ignitions and fire perimeters
was forced through the origin as no fire perimeter can occur without an ignition and vice versa (Fig. S2).



The minimum mapping unit (MMU) was 200 ha in CLFD before 1997 (Stocks et al., 2002), and 405 ha in
ALFD before 1988 (French et al., 2015). To minimize uncertainties because of recent changes in the
mapping accuracy, we removed fires smaller than 200 ha from the CLFD and fires smaller than 405 ha
from the ALFD similar as in Scholten et al. (2021b) and Veraverbeke et al. (2017). Similarly, ABoVE-
FED fires smaller than MMUs were excluded when developing these relationships. We used the
established statistical relationship between ignitions and fire perimeters in each ecoregion from 2001 to
2019 to estimate the annual numbers of ignitions between 1980 and 2000.

*2.4 Influence of snowmelt timing on ignition timing and fire size*
For each ignition location, we retrieved the snowmelt timing by averaging the MODIS-derived day of
snowmelt timing over each ignition location, including its spatial uncertainty derived from the ignition
algorithm. Snowmelt timing may be an important modulator of fire ignitions in the early fire season,
whereas seasonal soil moisture dynamics may more importantly influence fire behavior later in the fire
season (Flannigan et al., 2016; Gergel et al., 2017). To evaluate the relationship between snowmelt and
ignition timing between 2001 and 2019, we focused on ignitions that occurred early in the fire season. To
define early fire season ignitions, we first evaluated the correlation between the annual snowmelt timing
and ignition timing for all ignitions, per ecoregion. We then re-evaluated these relationships by only
including a fraction of the ignitions. This fraction was derived from taking a percentile of the ignition
timing distribution, between the first and 99th percentile. We generally found significant positive
correlations between snowmelt timing and ignition timing for all percentiles with a general decline in
correlation strength with inclusion of ignitions later in the fire season (Fig. S3). Thus, we set the ignition
timing threshold to the annual 20th percentile of the ignition timing distribution. For this threshold, all
ecoregions showed strong significant Pearson r correlation (range: 0.25 to 0.77) between snowmelt and
ignition timing (Fig. S3). By doing so, we retained 3 849 ignitions that occurred between the Julian days
58 and 294 across the study domain (Fig. S4).



We also compared all early- versus late-season ignitions to examine the importance of ignition
timing on fire size. The burned area caused by an ignition was assigned to the given day of the ignition. In
case of multiple ignition locations detected for one fire perimeter (approximately 4 % of the perimeters),
the burned area was assigned to the earliest ignition day of the year. We summed up the total burned area
between 2001 and 2019 per ignition day. The threshold between early and late ignition timing was again
set as the annual 20[th] percentile day of ignition timing per ecoregion.

*2.5 Climatic drivers of snowmelt and ignition timing*
The meteorological drivers of snowmelt timing and ignition timing were assessed with hourly
meteorological data derived from the fifth generation of the European Centre for Medium-Range Weather
Forecast's (ECMWF) reanalysis for the climate and weather (ERA5 reanalysis) at 0.25° resolution
(Hersbach et al., 2020). Fire weather data were collected from the Global ECMWF Fire Forecast ERA5
reanalysis dataset (GEFF-ERA5) of fire danger at 0.25° resolution (Vitolo et al., 2020). We extracted
convective potential available energy (CAPE), total precipitation, precipitation type (rain vs. snow), air
temperature at 2 m, and dewpoint temperature at 2 m from the ERA5 reanalysis. From these variables we
further derived relative humidity (Table S3). The fine fuel moisture code (FFMC), duff moisture code
(DMC), and drought code (DC) were collected from GEFF-ERA5 and are designed to represent the fuel
moisture of the top (1-2 cm organic layer, lag-time of 2/3 of a day), intermediate (5-10 cm sub-organic
layer, lag-time of 12 days) and deep (15-20 cm deep organic layer, lag-time of 52 days) soil layers (Van
Wagner, 1987). In regions regularly covered by snow, all fuel load variables are initiated on the third day
after the snow has melted while in regions without snow cover, calculations begin on the third
consecutive day with noon temperatures of < 12 °C (Lawson and Armitage, 2008). Here, we used the fire
weather variables as proxies for fuel dryness.
We calculated spatially explicit daily anomalies for all meteorological and fire weather variables
by subtracting the climatic daily averages between 1980 and 2019 from the daily observations between



2001 and 2019. We assessed the effect of precipitation, precipitation type (rain vs. snow), air temperature,
and relative humidity on snowmelt timing. Precipitation, temperature, and relative humidity anomalies
were averaged for the 30 days leading up to the day of snowmelt timing. The number of days with
snowfall, rainfall, and no precipitation were summed up for the 30 days leading up to the day of snowmelt
timing. The averages of all weather and fire weather anomalies, excluding precipitation type, between the
day of snowmelt timing and ignition timing were used to assess their influence on ignition timing.

*2.6 Temporal trends in snow-free season and fire season*
The temporal trends in the snow-free and fire season lengths were analyzed between 2001 and 2019. The
snow-free season length was calculatd by subtracting the ecoregion average day of snowmelt timing from
the ecoregion average day of snowmelt offset for each year from the MODIS product.

We evaluated several scenarios to define the fire season timing. For the fire season start, we

assessed scenarios between the day of the first ignition and the 20th percentile of the ignition timing
distribution. For the fire season end, we assessed scenarios between the day during which 80 to 99 % of
the annual burned area had occurred. First, we analyzed the percentage of annual burned area that was
excluded for different fire season start and ending scenarios (Fig. S5). We performed a sensitivity analysis
of the different cut-off values that showed no substantial changes in the relationship between the length of
the snow free period and the fire season length (Fig. S6). After evaluation, we chose the 1st percentile day
of ignition as fire season start and the day on which 99 percent of the annual burned area had occurred as
fire season end day. We subtracted the first day of ignition timing from the day of the 99th percentile total
burned area each year to calculate the fire season length. We also investigated changes in the snow-free
season length in relation to fire season length between 2001 and 2019.



*2.7 Statistical analysis*
All statistical analyses were performed in R statistical software version 4.2 (R Core Team, 2022). We
investigated temporal trends between 1980 and 2019 and between 2001 and 2019 in snowmelt timing
andthe number of ignitions using simple linear regression. The snow-free season length and fire season
length in each ecoregion were analyzed between 2001 and 2019 using simple linear regression. The
statistical difference in the average fire size between early and late ignitions was analyzed with a
Wilcoxon-Mann-Whitney rank sum test (Mann and Whitney, 1947). We distinguished between two
significance levels of $p < 0.05$ and $p < 0.1$.
To assess the ecoregional drivers of the divergent snowmelt timing and early-season ignition
timing, defined as the annual $20^{th}$ percentile ignitions, we used a linear mixed effect model. Prior to
testing, ignition locations in close proximity were spatially correlated (Moran's I = 0.30). We therefore
averaged all ignitions for each ecoregion per year to reduce the spatial autocorrelation. The snowmelt was
modeled as a function of weather while the ignition timing was modeled as functions of weather and fire
weather independently. This was to minimize the multi-collinearity in the generalized linear mixed-effect
models. We conducted our linear mixed-effect models with ecoregions (16 levels) as random effects
using the 'nlme' package (Pinheiro et al., 2022) (Tables S4 and S5) (eq. 1) to account for additional
temporal and spatial autocorrelation. We excluded year as random effect as it only explained around 3%
and 7% of the variation in snowmelt and ignition timing, respectively. We conducted three linear mixed-
effect models for all ecoregions combined ($n = 299$), ecoregions with earlier snowmelt timing trends ($n =$
186) and later snowmelt timing trends ($n = 113$) based on the MODIS-derived snowmelt timing (Table
S1):
$$y = X\beta + Zu + \varepsilon$$

(1)

where y is the response variable, $X\beta$ represents the fixed effects, where $X$ is a matrix of observed values
per variable and $\beta$ represents the regression coefficient for each variable. The $Zu$ term represents the



random effects, where $Z$ is a matrix for observed values per covariate of random effects and $u$ is the
random effect of the covariates. The error term $\varepsilon$ represents the residuals.
All variables were standardized prior to testing and the analysis was conducted for ignitions
between 2001 and 2019. The significance of the fixed effects was tested using likelihood ratio tests of the
reduced and full models. We used the Akaike information criterion (AIC) to verify the significance of the
models compared to reduced models (Zuur et al., 2009). The best model fit was chosen to be the linear-
mixed model with different intercepts per random effect (ecoregion) bur similar slopes for every predictor
and random effect. For further variable selection for our piecewise Structural Equation Model (pSEM),
we evaluated the influence of meteorological variables (Table S6) on the day of snowmelt timing and the
additional influence of snowmelt timing and the fuel codes on ignition timing through a redundancy
analysis in the R package 'vegan' (Oksanen et al., 2013) (Table S6). The significance of the unique
contribution of all drivers included in the two variance partitioning analyses was determined by adjusted
$R^2$ and $p < 0.05$. The shared variance and the residual variance between drivers were also computed
(Table S6).
We expected that the interactions between predictor variables and the snowmelt and ignition
timing constituted a complex network and therefore deployed a pSEM in the package 'piecewiseSEM'
(Lefcheck, 2016). The pSEM creates a single causal network from our deployed multiple linear-mixed
effect models that incorporates a random structure (Shipley, 2009). We included explanatory variables
linked to snowmelt and ignition timing based on analysis of bivariate relationships of meteorological and
fire weather data that could influence the timing of snowmelt and ignition. Bivariate relationships were
evaluated by simple linear regressions between snowmelt timing and the respective predictor variables,
and ignition timing at its potential explanatory variables (Table S7). The hypothesized network of
interactions in our pSEM was modelled for three individual pSEMs to test this hypothesized model of
interaction between weather, fire weather and snowmelt timing but also to describe the potential effect of
divergent snowmelt timing across the study domain. We modelled a pSEM: (1) for all ecoregions, (2)



ecoregions with early snowmelt timing trends in accordance with the MODIS trend analysis (Table S1),
and (3) ecoregions with later snowmelt timing trends in accordance with the MODIS trend analysis
(Table S1).

For modelling snowmelt timing, we hypothesized that, (1) as the total amount of precipitation

decreases and the air at the surface becomes drier, increased surface air temperature would accelerate
snowmelt timing. We also hypothesized that, (2) snowmelt timing would occur earlier with increased
days of no precipitation (smaller snowpack) or days with rain-on-snow events (more rainfall) compared to
snow-on-snow events (more snowfall). We hypothesized that, (3) earlier snowmelt timing would result in
earlier ignition timing. For modelling the influence of snowmelt timing on weather and fire weather
variables, we hypothesized that, (4) surface relative humidity and precipitation would decrease and limit
the evaporative cooling and in turn result in higher air temperatures. This would increase atmospheric
instability and the CAPE and would all increase the likelihood of earlier ignitions. Lastly, we
hypothesized that (5) earlier snowmelt timing would promote drying of fuels (FFMC, DMC, and DC)
more pronouncedly in ecoregions with earlier snowmelt timing. We allow for links between weather and
fire weather variables, since DC, DMC, and FFMC are derived from precipitation, relative humidity and
temperature while the calculation of FFMC also ingested wind speed. These interactions are included to
comply to statistical requirements of inclusion of missing paths in the pSEM analysis but left out of the
figure for simplicity reasons (Fig. S7). As the pSEMs can consist of many different linear models, we
fitted each component of the pSEM with a linear mixed-effect model. We assessed potential additional
variable interaction and their conditional independence using Shipley's test of dependence separation ($d$-
sep). The test is founded on the $\chi^2$ distributed and combines the Fisher's C statistics with $2j$ degrees of
freedom, where $j$ is the number of independent interactions in a basis set (Shipley, 2009) (eq. 2):
$$C = -2 \sum_{i=1}^{k} \ln(p_i)$$

(2)

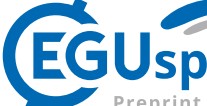



where $k$ is the number of independence claims, $p_i$ is the null probability of the independence test
associated with the $i^{th}$ independence claim.
The missing paths determined by the $d$-sep test were included in the hypothesized pSEM to accurately
analyze the network of dependent variables in our overall pSEM. The global goodness-of-fit of our
models and the hypothesized model was evaluated by the $d$-sep. With $p$-values $> 0.05$, the representative
model misses no paths and is in accordance with the hypothesized model (Shipley, 2009). The estimates
of paths from predictor variables to response variables for each pSEM were standardized for comparison
of effects across multiple responses and their indirect and total effects. The standardization of coefficients
was done by the ratio of the standard deviation of the independent and dependent variable of the given
variables (eq. 3):
$$\beta_{std} = \beta \times \left(\frac{sd_x}{sd_y}\right)$$

(3)

,where $\beta$ is the unstandardized coefficient, $sd_x$ is the standard deviation of the independent variable, and
$sd_y$ is the standard deviation of the dependent variable. The explained variation of snowmelt and ignition
timing from the different components in the pSEMs were analyzed using the marginal and conditional $R^2$.
Marginal $R^2$ represents the variation explained only by the fixed effects, and conditional $R^2$ shows the
variation explained by a combination of fixed and random effects.

**3. Results**
*3.1 Trends in snowmelt timing and ignitions*
The long-term (1980-2019) and short-term (2001-2019) snowmelt timing trends over boreal North
America showed somewhat similar patterns. Long-term snowmelt timing trends occurred earlier in the
northern ecoregions, however more pronounced in Northwestern boreal North America, with only interior
southern ecoregions showing later snowmelt timing between 1980 and 2019 (Fig. 2a). The spatial



divergence, however, promoted to a distinct west-east divergence in the snowmelt timing trend across
boreal North America, with increasingly earlier snowmelt observed in western boreal North America
versus later snowmelt in the eastern ecoregions between 2001 and 2019. This trend has also become more
pronounced in the last two decades (Figs. 1a and 2, and Table S1). The west-east divergence in snowmelt
timing ranged from advances of up to 11 days per decade in the western ecoregions to delays of up to 8
days per decade in the eastern part of the study region in the MODIS era (2001-2019). The long-term
NSIDC snow product (1980-2019) showed trends between advances in snowmelt timing of 3 days per
decade in the west to delays of 2 days per decade in the East (Table S1). On average, snowmelt advanced
1.6 (standard deviation: 0.7) days per decade ($p < 0.05$) in the western ecoregions (Fig. 2 A-H, J, L),
while snowmelt occurred 1.8 (standard deviation: 0.9) days per decade later in the central and eastern
ecoregions ($p = 0.05$) (Fig. 2 I, K, M-P). We observed the most pronounced earlier snowmelt trends of 2.1
(standard deviation: 0.5) days earlier snowmelt per decade ($p < 0.05$) in northwestern ecoregions (Fig. 2
A-E), while the most pronounced later snowmelt trends mainly occurred in the southern ecoregions of 1.1
(standard deviation: 0.8) days per decade ($p = 0.05$) (Fig. 2 M-P). The spatially diverging trends in
snowmelt timing are associated with similar trends in early spring (February-April) air temperature
between 1980 and 2019 (Fig. S8a). The northernmost ecoregions showed the largest increase in early
spring air temperature, while the southern ecoregions experienced decreasing early spring air temperature
over the last four decades. Superimposed on this north-south gradient, we also found that the west of the
study domain experienced pronounced early spring warming while the east of the study domain
experienced early spring cooling (Fig. S8). The distinct spatial divergence in short-term snowmelt timing
trends also follows a more pronounced short-term early spring air temperature dipole. Early spring time
air temperatures increased with up to 3.5°C over western ecoregions with earlier snowmelt timing trends
and decreased with up to 2.1°C over southeastern ecoregions showing delayed snowmelt timing (2001-
2019) (Fig. S8b).



In accordance with the spatial diverging trends in snowmelt timing and early spring air
temperatures, the trends in the number of ignitions also showed a west-east divergence. The northwestern
ecoregions that displayed a pronounced earlier snowmelt also exhibited an increase in the number of total
ignitions of $0.9 \times 10^{-6}$ (standard deviation: $0.8 \times 10^{-6}$) km$^{-2}$ decade$^{-1}$ ($p < 0.05$) (Fig. 2 A-G) between 1980
and 2019. The southwestern ecoregions of the Cold Deserts, Marine West Coast Forest, and Western
Cordillera demonstrated the strongest increasing trends in ignitions ($6.4 \times 10^{-6}$, standard deviation: $4.4 \times$
$10^{-6}$ ignitions km$^{-2}$ decade$^{-1}$, $p < 0.05$) (Fig. 2 H, J, L), while the central and eastern ecoregions showed an
overall decrease of $0.2 \times 10^{-6}$ (standard deviation: $0.3 \times 10^{-6}$) ignitions km$^{-2}$ decade$^{-1}$ ($p = 0.51$) (Fig. 2 I, K,
M-P). However, there was no spatially divergent trend in the temporal changes in ignition timing between
2001 and 2019. In 12 of 16 ecoregions, there was a shift towards earlier ignitions, when we included all
ignitions, with 7 ecoregions showing significantly earlier ignitions. The trends towards earlier ignition
ranged between 0.4 and 25 days per decade (Table S1). Of the four ecoregions that showed later ignition
timing trends, three were located in the southwest of the study domain (Boreal Plain, Cold Deserts, and
Western Cordillera), while the Western Taiga Shield was the only northern ecoregion with later ignition
timing (Fig. S9 and Table S1).

*3.2 Relationships between snowmelt and ignition timing*
In all ecoregions, we found significant positive relationships between snowmelt and ignition timing in the
early fire season (20$^{th}$ percentile of the ignition timing distribution) between 2001 and 2019 ($p < 0.1$) (Fig.
3). The strength of the relationships was similar across boreal North America and the advance in ignition
timing ranged between 0.22 and 1.43 days per day of earlier snowmelt (Fig. 3). Ignitions occurred later
and in a narrower temporal window in the northern ecoregions (Fig. 3 A-I, K) compared to the southern
ecoregions. Southern ecoregions also showed a more variable ignition timing at the beginning of the fire
season (Fig. 3 J, L-P). Furthermore, the southwestern ecoregions of our study domain showed a bimodal
ignition timing distribution, which could point to differences in ignition cause. Anthropogenic ignitions





dominate earlier in the fire season while lightning ignition are more prevalent around the summer solstice
(Fig. 3 J, L). Nonetheless, when we separated the anthropogenic and lightning ignitions, and ignitions
with unknown cause, we still observed positive relationships between snowmelt and ignition timing for
all causes (Table S8).

*3.3 Trends in snow-free and fire season lengths*
The temporal changes in the snow-free season length and the fire season length also showed a distinct
west-east divergence. Corresponding to the overall trends in the snowmelt timing, we found that the
northwestern ecoregions that show increasingly earlier snowmelt also experience a prolonged snow-free
season of 7.1 (standard deviation: 4.2) days per decade ($p < 0.1$) (Figs. 2a A-H, J, L and 4a A-H, J, L)
between 2001 and 2019. The southeastern ecoregions where snowmelt was increasingly occurring later in
spring exhibited a shortening of the snow-free season of 7.3 (standard deviation: 4.7) days per decade
(Figs. 2a I, K, M-P and 4a I, K, M-P), however not significant ($p = 0.12$), between 2001 and 2019. The
positive trend in snow-free season length was significant in 5 of the 16 ecoregions, while only the Eastern
Taiga Shield showed significant shortening trend in snow-free season length between 2001 and 2019 ($p <$
0.1) (Table S9). We observed similar spatial divergence in the long-term trends in changes in the snow-
free season length between 1980 and 2019 (Fig. S10).

The temporal changes in fire season length showed a west-east gradient in complement to a

north-south gradient for our study domain (Fig. 4b). The fire season length between 2001 and 2019
increased substantially from 1.7 and up to 25.3 days per decade and on average 5.8 (standard deviation:
7.6) days per decade for the northern ecoregions except in Taiga Plain (Fig. 4b A-H, K and Table S9) ($p =$
0.45). The southern ecoregions experienced an average shortening of the fire season length between 2001
and 2019 of 18.2 (standard deviation: 10.5) days per decade (Fig. 4b, I, J, L, M-O) ($p < 0.1$). The
northernmost ecoregions in our study region have experienced the largest prolonging of the fire season
over the last two decades of 18.0 (standard deviation: 10.1) days per decade (Fig. 4b, B, C, G) ($p < 0.1$).



429   We found that the snow-free season and fire season lengths between 2001 and 2019 were highly

430  correlated (Fig. 4c). There was a consistent significant positive relationship between the snow-free season

431  and fire seasons lengths across boreal North America between 2001 and 2019 regardless of the thresholds

432  set for the fire season start and end (Fig. S5). Across the study domain, we observed a lengthening of the

433  fire season of 1.7 days for every day of prolonged snow-free season. The length of both the snow-free

434  season and the fire season was shortest in the northern ecoregions and gradually prolonged for more

435  southern ecoregions (Fig. 4bc). We also found that the trends in snow-free and fire season length tend to

436  correlate positively with each other with a prolonging of the fire season of 0.9 days per decade for every

437  day per decade increase in the snow-free season ($p < 0.05$) (Fig. 4d).

439  *3.4 Ignition timing and fire size*

440  Early-season ignitions resulted in significantly larger fires than late-season ignitions in 8 out of the 16

441  ecoregions ($p < 0.1$) (Fig. 5). Only in two ecoregions, Alaska Tundra and Eastern Softwood Shield, late

442  season fires on average grew larger compared to early season fires ($p = 0.58$ and $p = 0.76$, respectively)

443  (Fig. 5 B,P). On average, ecoregional early season fires grew between 30 and 600 % larger than

444  ecoregional late season fires, while early season fires grew 77 % larger than late season fires across the

445  whole study region (Fig. 5). The relative increase in fire size from early season fires compared to late

446  season fires was more pronounced in southern ecoregions than in northern ecoregions (Table S9). Alaska

447  Boreal Interior, Taiga Plain, and Western Taiga Shield experienced the largest early season fires (23 218

448  (standard deviation: 7 557) ha) compared to the other ecoregions (9 922 (standard deviation: 5 192) ha)).

449  However, in these ecoregions, early-season fires accounted for approximately one third of the total burned

450  area whereas in the southern ecoregions early-season fires accounted for more than half of the total

451  burned area (Fig. 5 J, L, O and Table S9). Across our study domain, the 20[th] percentile earliest ignited

452  fires accounted for an average of 40.6 (standard deviation: 14.2) % of the total annual burned area (Table

453  S9).



*3.5 Influence of snowmelt timing on ignition timing*

The pSEM for all ecoregions matched reasonably well with our hypothesized pSEM model (Fisher's $C_{80}$

= 82.24, $p$ = 0.41; Fig. 6) and explained 38 % of the variation in the snowmelt timing (marginal $R^2$ (M-

$R^2$) = 0.38, conditional $R^2$ (C-$R^2$) = 0.50) and 48 % of the variation in ignition timing (M-$R^2$ = 0.35, C-$R^2$

= 0.35) (Fig. 6). The model fits for ecoregions with earlier snowmelt timing trend (Fisher's $C_{86}$ = 96.31, $p$

= 0.21) and later snowmelt timing trends (Fisher's $C_{112}$ = 107.14, $p$ = 0.61) were poorer than the pSEM fit

on all ecoregions (Fig. S7). Nonetheless, the variance explained in the snowmelt timing and ignition

timing were generally better when splitting ecoregions between those with earlier snowmelt trends

(snowmelt: M-$R^2$ = 0.32, C-$R^2$ = 0.32, ignition: M-$R^2$ = 0.54, C-$R^2$ = 0.54) and later snowmelt trends

(snowmelt: M-$R^2$ = 0.53, C-$R^2$ = 0.53, ignition: M-$R^2$ = 0.53, C-$R^2$ = 0.55) (Fig. S7).

These results show that snowmelt timing was driven by air temperature, without significant

influence of precipitation type and amount and humidity. The earlier snowmelt timing was correlated with

high anomalies in air temperature, while the air temperature was generally lower than the climatological

averages with later snowmelt timing (Tables S10-S12). The pSEM model results also show that earlier

snowmelt timing promoted fuel drying across ecoregions (Fig. 6 and Table S10).

Snowmelt timing itself had the strongest individual influence on ignition timing across all

ecoregions and models also after accounting for weather and fire weather. The cascading effect of

accelerated drying of organic soils from earlier snowmelt timing carried over to the timing of ignition. For

all models, the DMC had the strongest influence on the ignition timing, while the FFMC significantly

affected ignition timing across all ecoregions and over the ecoregions exhibiting earlier snowmelt timing

(Fig. 6, and Fig. S7a). For ecoregions with later snowmelt trends, only the slow responding fuel moisture

codes (DMC and DC) significantly influenced the timing of ignition. For ecoregions with earlier

snowmelt timing, DC influenced the ignition timing positively and earlier ignitions generally occurred



under wetter DC conditions. The fuel moisture codes together more strongly influenced ignition timing
compared to snowmelt timing and weather variables across models (Tables S10-S12).

Early snowmelt may also affect larger-scale atmospheric dynamics. We found that earlier

snowmelt timing contributed to the destabilization of the atmosphere through increased convective
available potential energy (CAPE) across ecoregions (Fig. 6), in particular for ecoregions with earlier
snowmelt timing (Fig. S7a). Early snowmelt was associated with higher temperatures and lower humidity
in the overall model. These favorable weather conditions led to earlier ignition in the overall model and
the model for ecoregions with earlier snowmelt timing. Early ignitions were associated with lower
relative humidity and higher air temperatures driven by the earlier snowmelt timing (Fig. 6 and Fig. S7a).
Snowmelt timing itself had the strongest individual influence on ignition timing across all ecoregions and
models.

**4. Discussion**
*4.1 Diverging spatial trends in snowmelt timing and ignitions*
We found the co-occurrence of a pronounced continental dipole in decadal trends of snowmelt timing and
number of fire ignitions across arctic-boreal North America. We observed increasingly earlier snowmelt
and an increase in the number of fire ignitions in northwestern boreal North America between 1980 and
2019. In contrast, snowmelt timing has simultaneously been occurring later and the number of fire
ignition decreased in the last decades in the southeastern part of our study domain. The changes in
snowmelt timing that we found in our study are corroborated by earlier work demonstrating both
increasing and decreasing trends in snow-cover over southeastern and northwestern boreal North
America, respectively (Chen et al., 2016). Furthermore, Bormann et al. (2018) found an earlier onset of
spring snowmelt in northwestern boreal North America in contrast to later snowmelt or no changes in
snowmelt timing over southeastern boreal North America. We also found that the west-east diverging



trend in snowmelt timing has become more pronounced in the last two decades compared to the longer-
term trend since 1980. These observations followed the divergent trend of less pronounced changes in
long-term early spring air temperature (1980-2019) and distinct dipoles in early spring air temperature
over boreal North America between 2001 and 2019 (Fig. S8). Similar to Cohen et al. (2014), we found
small changes in air temperature between 1980 and 2019 in the northern and southern parts of our study
domain (Fig. S8a). The last two decades of enhanced west-east divergence in snowmelt timing followed
the development of a pronounced west-east dipole in early spring air temperature as observed in our
linear-mixed effect models (Table S4) and also observed in two consecutive recent winters between 2013
and 2015 (Singh et al., 2016). As higher early spring temperatures promote earlier snowmelt, the snow-
albedo feedback will in turn result in higher temperatures (Déry and Brown, 2007). In this way, the
presence of a dipole of changes in early spring air temperature and snowmelt timing over boreal North
America might indicate that both processes enforce each other on sub-continental scales.

Besides regional changes in early spring air temperature, large-scale atmospheric dynamics may

also have influenced snowmelt timing and the number of ignitions as observed in our study (Cohen et al.,
2014; Zhao et al., 2022). Changes in sea ice and snow cover (Zou et al., 2021) may have large impact on
the location of the polar jet stream and tropospheric ridge persistency causing temperature extremes
(Francis and Vavrus, 2012; Kim et al., 2014; Horton et al., 2016). In recent decades, these persistent
tropospheric ridge patterns have been located over the northwestern part of our study domain which traps
and slows the progression of Rossby waves eastwards (Francis and Vavrus, 2012; Jain and Flannigan,
2021) resulting in downstream troughing over the east (Singh et al., 2016). This tropospheric ridge leads
to a blocked anticyclone in the west, causing higher air temperatures and increased burned area, and an
associated cyclone over eastern North America with lower temperatures and less burned area (Skinner et
al., 1999; Cohen et al., 2014; Sharma et al., 2022). Further, the stratospheric vortex, westerly winds
formed in the stratosphere during winter time, may have weakened and consequently sudden stratospheric
warming (SSW), rapid heating in the stratosphere over the North Pole, have caused winter cold-spells



over eastern Canada over the last four decades (Kretschmer et al., 2018b). The effect of winter cold-spells
may carry over into spring delaying the snowmelt timing and thus the fire season. The presence of a
dipole in snowmelt timing and ignition trends in our study is likely related to: (1) changes in the
stratospheric vortex and SSW that send winter cold-spells over the eastern part of the study domain
(Kretschmer et al., 2018a) and as a consequence annual mean air temperature anomalies divergence from
increasing in the west to decreasing in the east of boreal North America in the last decades (Cohen et al.,
2012; Coumou et al., 2018). (2) Changes in the location of the summer jet as a consequence of longer
persistence of positive geopotential anomalies over the western part of our study domain (Jain and
Flannigan, 2021; Zou et al., 2021). However, the persistent ridge formation could possibly also be a result
of the divergent snowmelt trend caused by the SSW, Both the soil moisture and albedo feedback between
snowmelt timing and temperature may have further strengthening the diverging trends. In our study, these
atmospheric processes and soil moisture feedbacks may also have led to the enhanced fuel dryness in
western ecoregions that has driven the large increases in number of ignitions compared to the other
ecoregions (Abatzoglou and Williams, 2016; Holden et al., 2018).

*4.2 Influence of snowmelt timing on ignition timing and fire size*
By focusing on the start of the fire season, we were able to disentangle the effect of snowmelt timing on
ignition timing. Previous studies found no significant effects of snowmelt timing on annual burned area,
with snowmelt timing being regarded as a minor driver of annual burned area compared to meteorological
variables (Jolly et al., 2015; Kitzberger et al., 2017). Nonetheless, snowmelt timing has shown to play a
crucial role in altering fuel dryness and the frequency of large fires over a temperate forest in the western
United States (McCammon, 1976; Westerling et al., 2006). Our results show that snowmelt timing has a
strong influence on early ignitions in all ecoregions of boreal North America. This relationship
diminished when snowmelt timing was compared to progressively later ignitions (Fig. S3). This may be
due to the importance of the spring window, the period between snowmelt timing and leaf flush, on early-





season fires (Parisien et al., 2023). During the spring window deciduous and mixed forests are very
conductive to fire and ecoregions experience the longest spring window corresponded to where we also
found the highest early fire ignition density (Fig. 5J, L-M) (Parisien et al., 2023). Also, the longest spring
window was found in the interior west of Canada (Parisien et al., 2023), which coincides with the
ecoregions with most fire ignitions observed in our study (Fig. 1). Late-season ignitions in July, August
and September may be more influenced by long-term drought and synoptic weather conditions than by
snowmelt timing (Jain et al., 2017; Holden et al., 2018).

We find that throughout boreal North America, fires caused by early season ignitions following

earlier snowmelt also on average grew larger than fires ignited in the late fire season. This was in
accordance with earlier findings limited to Alaska, USA, and the Canadian Northwest Territories
(Veraverbeke et al., 2017). Because of the early snowmelt and the earlier ignition timing, early season
fires have longer temporal windows with potential for favorable warm and dry weather conditions
conductive to fire spread (Sedano and Randerson, 2014). Indeed, the 20 % earliest ignitions resulted in
approximately 40 % of the total burned area across the study domain between 2001 and 2019. In the
future, the contribution of early fires to burned area might increase with warmer and drier weather
conditions leading to earlier snowmelt and thus an increased likelihood for earlier and larger fires over
boreal North America (Flannigan et al., 2005, 2013).

*4.3 Changes in the snow-free and fire season lengths*
We found a north-south gradient in the changes in the actual fire season length ranging from a prolonging
of 30 days per decade in northern ecoregions to a shortening of 25 days per decade in southern
ecoregions. Previous studies have mainly found the prolonging of the potential fire season to be between
3 and 30 days per decade over boreal North America (Wotton and Flannigan, 1993; Flannigan et al.,
2013; Jolly et al., 2015; Jain et al., 2017). These estimates of the prolonging of the potential fire season



were based on changes in fire weather (Flannigan et al., 2013; Jain et al., 2017). Other studies have, with
the usage of governmental fire perimeter data, also found a prolonging of the fire season length limited to
the western North America, Alberta and Ontario in Canada (Westerling et al., 2006; Albert-Green et al.,
2013). In our study, we used daily fire spread data from spaceborne data to determine fire start and end
dates (Skakun et al., 2021; Potter et al., 2023). This approach, however, relies on MODIS active fire data
and therefore is limited to the MODIS era. Longer-term accurate temporal and spatial data on ignition
timing and end of burning is needed to assess the changes in the actual fire season on a climatic timescale
and a continental scale. Our results suggest that a change in the duration of the snow-free season is almost
one to one related to a change in the duration of the fire season across boreal and arctic North America.
However, the effect may be of more importance on the fire season start than the end of the fire season as
this is often marked by the first rainfall in autumn in adequate amounts for extinguishing fires and
rewetting dried out fuels preventing new ignitions. Climate change induced changes in the amount and
timing of autumn rainfall will likely effect the timing of the fire season end (Holden et al., 2018; Goss et
al., 2020). Although, recent studies also showed that some fires overwinter and re-emerge the following
spring (McCarty et al., 2020; Scholten et al., 2021b; Xu et al., 2022) challenging the concept of a
demarcated fire season. In a warmer North American Arctic-boreal region, the snow-free season will
likely prolong with a consequent lengthening of the fire season, both starting earlier in spring and
prolonging later into autumn (Flannigan et al., 2013).

*4.4 Cascading effects of snowmelt timing on weather and ignition timing*
In the three piecewise structural equation models (pSEMs), anomalies in snowmelt timing were only
attributed to anomalies in air temperature (hypothesis 1). Our models did not confirm the importance of
the amount or the type of precipitation for snowmelt timing observed in previous research (hypothesis 2)
(Barnett et al., 2005; McCabe et al., 2007). However, air temperature also affected precipitation types in
our models which, although statistically insignificant, showed divergent influences on snowmelt timing



between ecoregions with earlier and later snowmelt trends (Tables S11 and S12). This suggests that the

air temperature dipole observed in the last two decades (Fig. S8) may influence precipitation, including

snowpack volume and persistency (Brown and Mote, 2009) and therefore likely also snowmelt timing

(Barnett et al., 2005). Nonetheless, snowmelt timing itself largely influenced ignition timing regardless of

ignition source and ecoregion, and we additionally found snowmelt timing to be an important early

indicator for early season fires in the North American Arctic-boreal region (hypothesis 3). We also found

a cascading effect of snowmelt timing on meteorological conditions that carried over into the influence on

ignition timing. Relationships between warm and dry conditions and ignitions and fire spread have been

established before (Sedano and Randerson, 2014; Veraverbeke et al., 2017). This was only apparent for

the overall model and the model including ecoregions with earlier snowmelt timing (hypothesis 4). This

suggests that land-atmosphere dynamics are altered by changes in snowmelt timing as it influences the

soil moisture content which is proportional to evapotranspiration changing the land energy balance

(Seneviratne et al., 2010). Also, this in combination with anomalously high springtime temperatures

promoted greening of vegetation and desiccation of soils in other boreal regions changing the impact of

the warming in the atmospheric (Gloege et al., 2022). These land-atmosphere dynamics may have been

potential pathways for extreme fire seasons in Siberia (Scholten et al., 2022), and our pSEM results

indicate that similar dynamics may be in place over ecoregions with earlier snowmelt timing. The

ecoregions with later snowmelt timing, which did not show this carry-over effect of snowmelt timing on

weather and fuel moisture to ignition, also corresponded to the more densely populated regions. This may

be due to the elevated potential for anthropogenic ignitions that again coincide with the more flammable

vegetation during the spring window (Wotton et al., 2010; Parisien et al., 2023) and less with favorable

weather conditions compared to lightning ignitions.

The results of our study also point to cascading effects of changes in snowmelt timing on dry fuel

availability that carried over into the ignition timing across all models. The fine fuel moisture and duff

moisture codes showed significant influences on ignition timing, while the drought code did not



(hypothesis 5). This is in agreement with previous studies that indicate that the ignition of fires in boreal
North America strongly depends on the immediate dryness of the fine fuels (Abatzoglou and Williams,
2016; Hessilt et al., 2022). The effects of earlier snowmelt timing on enhanced desiccation of fuels
observed in three forests sites (McCammon, 1976) may be broadly applicable across boreal North
America. As observed in our study, dry fuels can directly promote ignition timing as they are readily
ignitable (Hessilt et al., 2022), but this may also be indirect through the influence on aboveground
biomass senescence and ecosystem production (Liu et al., 2020).
Our pSEM analysis gives a simplified overview of relationships between snowmelt timing, land-
atmosphere dynamics, and fire ignitions. However, we acknowledge that these interactions are highly
coupled. The influence of snowmelt timing on atmospheric variables through surface albedo change and
altered soil moisture may be difficult to decouple from the atmospheric variables and their persistent
seasonal patterns on snowmelt timing itself. We therefore call for a better understanding of the role of
snowmelt timing on land-atmospheric dynamics affecting boreal fires. Specifically, large-scale influence
of continuous snowmelt on soil and fuel properties, e.g. soil and fuel moisture, and atmospheric
conditions e.g. vapor pressure deficit, and vice versa. Understanding these interactions and feedbacks
could further advance our comprehension of how climate change is affecting changing boreal fire
regimes.

**5. Conclusion**
We found a pronounced west-east divergence of recent changes in snowmelt timing and the number of
fire ignitions across boreal North America. Our results point to a clear trend of earlier spring snowmelt in
the northwestern ecoregions, while the southern and eastern ecoregions showed an increasingly later
snowmelt timing over the last decades. Similarly, the total number of fire ignitions increased in the
northern and western ecoregions, while the southeastern ecoregions experienced little to no changes in the



number of early fire ignitions. We conclude that climate warming resulted in increasingly earlier
snowmelt in north-western boreal North America, which in turn led to earlier fire ignitions, which tended
to grow into larger fires.

The temporal trends in snowmelt and ignitions across boreal North America followed the same

spatial pattern of temporal trends in early spring air temperature over the last four decades. Snowmelt and
ignition timing were positively correlated across all ecoregions and earlier snowmelt was the main driver
for earlier fire ignitions across all ecoregions. Further, we found a cascading effect of elevated air
temperature and earlier snowmelt that carried over into earlier drying of fuels, which resulted in earlier
ignitions across the study domain. This cascade was more pronounced over ecoregions with increasingly
earlier snowmelt timing than over those with increasingly later snowmelt timing. Our work points to the
important impact that snow cover and snowmelt timing have on fire ignitions and fire size across boreal
North America, as well as the influence of changes in snowmelt timing on changes in fire regimes. In a
warming North American boreal forest, earlier snowmelt will likely result in increasingly earlier and
larger fires.





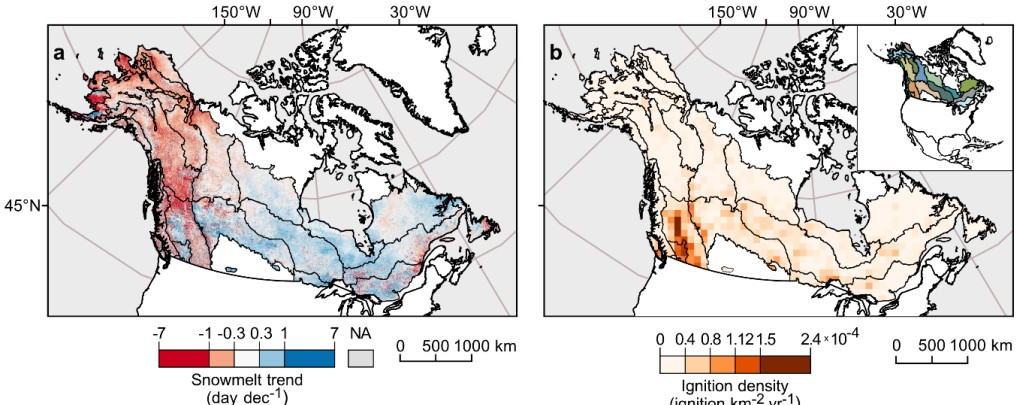


**Figure 1** (a) Trend in snowmelt timing between 2001 and 2019 derived from Moderate Resolution Imaging Spectroradiometer
(MODIS) for the study domain overlaid by second-level ecoregions (US EPA, 2015) and (b) the mean annual ignition density per
100 x 100 km grid cells between 2001 and 2019. All pixels exceeding the average pixel snowmelt timing ± 3 standard deviation
were excluded and set to not applicable (NA: grey).



**Figure 2** Trends in snowmelt timing and ignitions for all ecoregions (A-P) between 1980 and 2019 (a). The slope is given for all ecoregions, and its significance level is indicated by * ($p < 0.1$) or ** ($p < 0.05$). The magnitude and direction of the long-term trends in daily snowmelt timing (b) and number of ignitions (c) from 1980 to 2019.

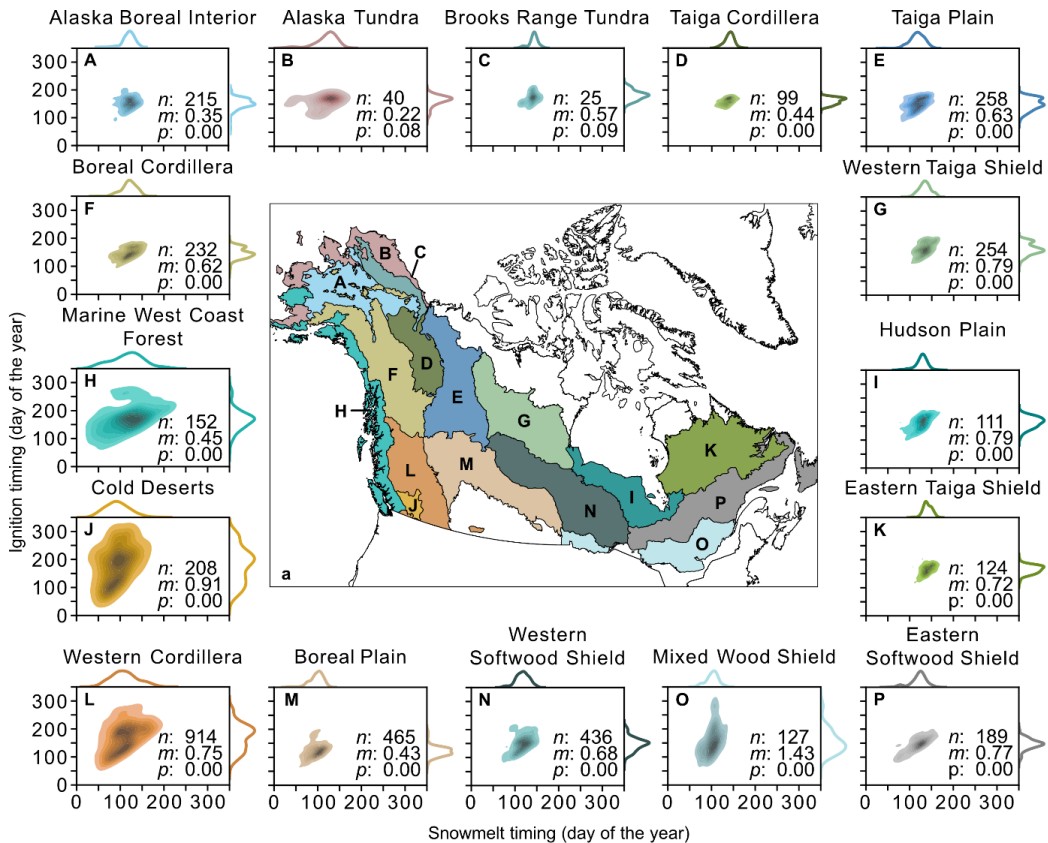

673

**Figure 3** Relationship between snowmelt and ignition timing for all ignitions of the annual 20th percentile of the ignitions timing
distribution per ecoregion, and their density plots (A-P). The number of ignitions ($n$), the slope ($m$), and the significance level ($p$)
are indicated for each ecoregion.



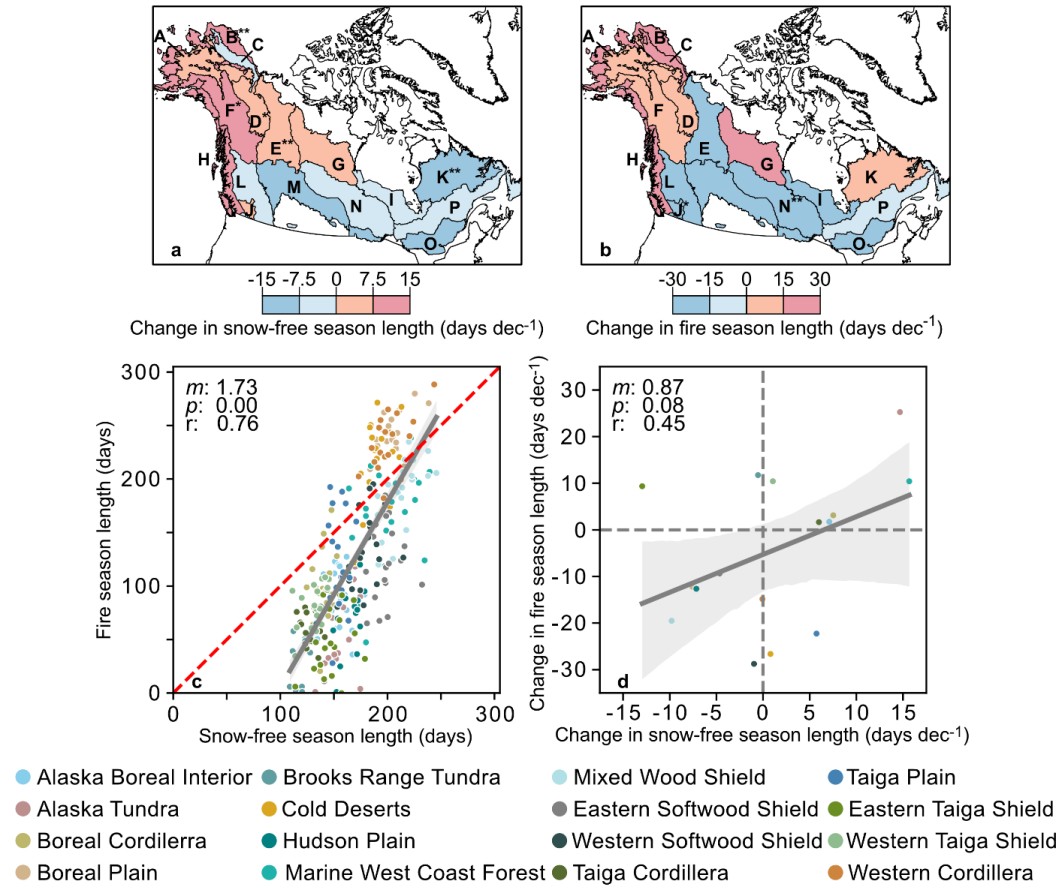

**Figure 4** Changes the snow-free season length (days decade$^{-1}$) for all ecoregions (A-P) between 2001 to 2019 (a), and the changes in the fire season length (days decade$^{-1}$) per ecoregion (A-P) between 2001 to 2019 (b) (Table S9). Letters correspond to the respective ecoregion names (Fig. 2) and significant relationships are indicated by * ($p < 0.1$) and ** ($p < 0.05$). The relationship between the annual absolute length of the snow-free season from the MODIS-product (days) and annual length of the fire season for all ecoregions (c), and the ecoregional trends in snow-free and fire season length (days dec$^{-1}$) (d).



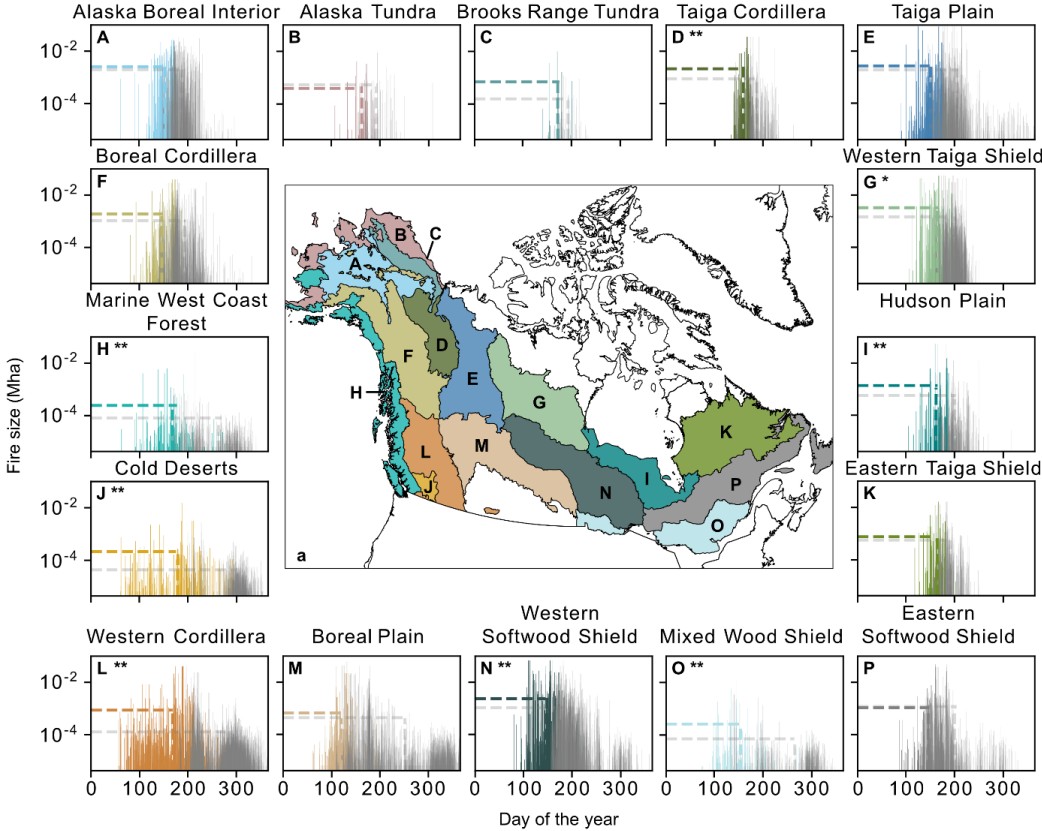

**Figure 5** Fire size as a function of ignition timing for all ecoregions (A-P). The 20th percentile day of ignition was set as threshold to discriminate between early and late season fires. The colored dashed lines indicate the mean ignition timing and fire size for all early season ignitions while the gray dashed lines indicate the mean ignition timing and fire size for all late season ignitions. Significant larger early season fires were indicated by * ($p < 0.1$) and ** ($p < 0.05$). Note the logarithmic scale for fire size.



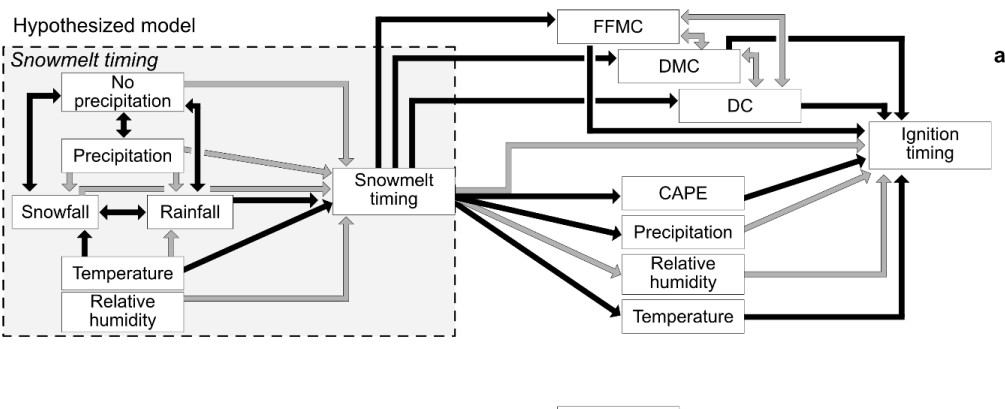

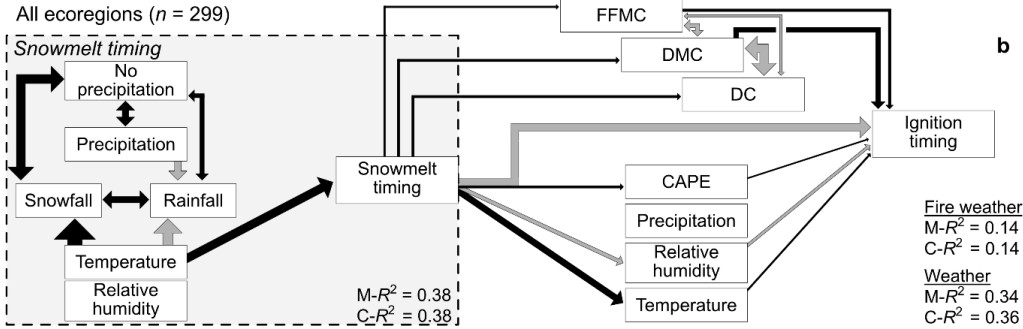

**Figure 6** Piecewise structural equation model (pSEM) of the hypothesized snowmelt and ignition timing model (a) and its fit for all ecoregions (b). Gray arrows represent positive effects and black arrows indicate negative effects. The single-headed arrows show significant direction of causal relationships, while double-headed arrows represent significant non-causal relationships ($p <$ 0.05). All arrows are scaled to their respective effect size (Table S10). Marginal $R^2$ (M-$R^2$) indicates the variation solely explained by the fixed effects and conditional $R^2$ (C-$R^2$) represents the variation explained by both the fixed and random effects.



**Code availability**

Code is available upon request from the corresponding author.

**Data availability**

The MODIS and NSIDC snow cover data is publicly available from the National Snow and Ice Data Center (MODIS: https://nsidc.org/data/mod10a1/versions/6, NSIDC: https://nsidc.org/data/nsidc-0046/versions/4). The burned area data is publicly available from the Oak Ridge National Laboratory Distributed Active Archive Center for Biogeochemical Dynamics (ORNL-DAAC) (https://doi.org/10.3334/ORNLDAAC/2063). Fire ignition data from Alaska, Yukon, and the Northwest Territories is available from the ORNL DAAC (https://doi.org/10.3334/ORNLDAAC/1812). The ignition data across boreal North America that we generated in this study is under the process to become freely available on the ORNL DAAC. All meteorological and fire weather variables were derived from the fifth generation of the European Centre for Medium- Range Weather Forecast; (meteorological variables:, https://cds.climate.copernicus.eu/cdsapp#!/dataset/reanalysis-era5-single-levels?tab=overview, and fire weather indices: https://cds.climate.copernicus.eu/cdsapp#!/dataset/cems-fire-historical?tab=overview).

**Supplementary information**

The supplement related to this article is available online at doi:

**Author contribution**

T.D. Hessilt designed the study with input from B.M. Rogers, R.C. Scholten and S. Veraverbeke. T.D. Hessilt performed the analyses and wrote the manuscript with inputs from all authors.

**Competing interests**

The authors declare no competing interests.

**Acknowledgements**
This work was carried out under the umbrella of the Netherlands Earth System Science Centre (NESSC). This project has received funding from the European Union's Horizon 2020 research and innovation programme under the Marie Skłodowska-Curie, Grant Agreement No. 847504. The contribution of R.C.S. was funded by the Dutch Research Council through Vidi grant 016.Vidi.189.070 awarded to S.V. The contribution of T.A.J. was funded by the European Research Council through a Consolidator grant under the European Union's Horizon 2020 research and innovation program (grant agreement No. 101000987) awarded to S.V. B.M.R. acknowledges funding from the NASA Arctic-Boreal Vulnerability Experiment (NNX15AU56A), the Gordon and Betty Moore Foundation (grant #8414), and funding catalyzed through the Audacious Project. T.D.H would like to thank D. Coumou for fruitful discussions on the effect of land-atmosphere dynamics related to snowmelt.



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
