# Peer review of "Geographically divergent trends in snow disappearance timing and fire ignitions across"

_EGUsphere, 2023_

## Author Comment (AC1)

**We thank the reviewer for helpful and in-depth comments. We will address those in a revised version and believe that this will make the manuscript stronger. In this file, the original comments are copied with our responses in bold underneath.**

Major Comments

The study by Zeng et al., titled "Geographically divergent trends in snowmelt 1 timing and fire ignitions across boreal North America," reported the influence of snowmelt timing on fire ignitions across the ecoregions of boreal North America. They found spatially divergent trends in early (late) snowmelt that led to an increasing (decreasing) number of ignitions in the northwestern (southeastern) ecoregions between 1980 and 2019. Early snowmelt is a proxy for early ignition but may also result in a cascade of effects from early desiccation of fuels and favorable weather conditions that lead to earlier ignition. This indicates that snowmelt timing is an important trigger for land–atmosphere dynamics.

Overall, this paper is logical and worthy of publication. However, minor revisions are required prior to publication.

Please note the following points.

1.  The major limitation of this paper was the incorrect use of the term "snowmelt." While snowmelt is also associated with snow cover, the authors must understand that the majority of snowmelt occurs at 100% snow cover. The term "snowmelt" used in this paper is incorrect and should be corrected throughout the paper, as it is more accurate than the terms "snow disappearance date" or "snow disappearance timing." Otherwise, the readers may fail to understand the authors' analyses. I believe that the title also needs to be revised.

    **We appreciate the thorough explanation and we will use the term snow disappearance in the revised manuscript.**

2.  The authors used surface data from ERA5 as the climate drivers of snowmelt and ignition timing, but ERA5 is a model estimate, and surface data are known to have bias. The bias is particularly large at high latitudes because of the lack of land-based observations and thus differs from ground-based data. It is necessary to demonstrate the validity of using ERA5 in this study with a reference.

    **Thank you. We have added a paragraph in our method section demonstrating that the use of ERA5 reanalysis data is acceptable for high latitude regions. Line 226-227:**

    *"ERA5 reanalysis data have been used before in other studies that investigated extreme weather events and fires in the northern high latitudes (Gloege et al., 2022; Parisien et al., 2023). Furthermore, several of the ERA5 variables, such as precipitation, surface temperature, and specific humidity have been validated with ground observations over the study region (Alves et al., 2020)."*

    *Alves, M., Nadeau, D. F., Music, B., Anctil, F., & Parajuli, A. (2020). On the performance of the Canadian Land Surface Scheme driven by the ERA5 reanalysis over the Canadian boreal forest. Journal of Hydrometeorology, 21(6), 1383-1404.*

    *Gloege, L., Kornhuber, K., Skulovich, O., Pal, I., Zhou, S., Ciais, P., & Gentine, P. (2022). Land-Atmosphere Cascade Fueled the 2020 Siberian Heatwave. AGU Advances, 3(6), e2021AV000619.*

*Parisien, M. A., Barber, Q. E., Flannigan, M. D., & Jain, P. (2023). Broadleaf tree phenology and springtime wildfire occurrence in boreal Canada. Global Change Biology.*

3. (4) Surface relative humidity was used to model snowmelt timing, but I do not consider it as a good indicator of atmospheric dryness because surface relative humidity varies significantly depending on temperature. Instead, I recommend using the surface saturation deficit. The model in Figure 6 should also be recalculated using the saturation deficit because temperature and relative humidity vary almost identically, which is not desirable as a variable in a hypothesized model.

**The surface saturation deficit is in essence very similar to vapor pressure deficit that is often used in fire studies. Both variables quantify the difference between the available moisture in the air and the air's total moisture capacity at saturation. In response to your comments, we have considered an alternative model in which we modelled snow disappearance timing with vapor pressure deficit (VPD) rather than relative humidity (RH).**

[Figure]

**Fig. 1S The pSEM model as the initial model but relative humidity is substituted by vapor pressure deficit.**

**The overall model performance remained the same regardless of variable. The prediction of snow disappearance timing is similar using VPD or RH. The substitution of RH with VPD in prediction of ignition timing did not change the direction of convective available potential energy (CAPE), VPD, and temperature but slightly diminished their effect ($p > 0.05$). Only snow disappearance timing and precipitation remained significant contributors of earlier ignition timing ($p < 0.05$). Further, snow disappearance timing influenced VPD, similar as with RH. Part of this can be explained by the correlation between VPD and temperature as VPD was derived from RH and temperature. Therefore, we also ran a model in which temperature was excluded as it is also indirectly represented by the VPD.**

[Figure]

**Fig. 2S The piecewise structural equation model as the initial model but relative humidity and temperature are substituted by vapor pressure deficit.**

This resulted in similar results as our initial analysis, but the main difference is that RH and temperature were both represented through VPD.

We obtained similar results when modelling snow disappearance and ignition timing for ecoregions with earlier and later snowmelt timing when using VPD instead of RH (Figure S7 original manuscript).

[Figure]

**Fig. 3S The piecewise structural equation model for ecoregions with earlier and later snow disappearance timing as the initial models but relative humidity is substituted by vapor pressure deficit.**

The inclusion of VPD diminished the influence of other weather variables on ignition timing (Fig. 3S). The only difference was that VPD influenced ignition timing insignificantly (p > 0.05) compared to the model with RH in ecoregions with later snowmelt timing (Fig. S7b original manuscript). The diminished effect of other weather variables was caused by the effect of VPD. We also run the models where temperature is excluded (Fig. 4S), which gave similar results as we observed for the model for all ecoregions (Fig. 2S).

[Figure]

**Fig. 4S The piecewise structural equation model models for ecoregions with earlier and later snow disappearance timing as the initial models but relative humidity and temperature are substituted by vapor pressure deficit.**

In summary, we understand the reviewer's comment, but found that the inclusion of VPD did result in very similar model structure as the original models with RH, albeit with slightly lower performance. We will therefore keep our models with RH as the main analysis in the revised manuscript, and we will add the alternative models with VPD as supplementary figures.

4.  Figure 4d shows that there was more variation than a positive correlation, but we cannot say that there was a positive correlation. I would like to have this figure removed or a rationale added to say that the correlation was strong for those with large variations.

We agree that there is large variation in the data. We will keep the figure 4d but will include a paragraph describing the correlation and variation we observed in the data:

*"We also found that the trends in snow-free and fire season length tended to correlate positively with each other with a prolonging of the fire season of 0.9 days per decade for every day per decade increase in the snow-free season (p = 0.08) with large variation between ecoregions (the trends in*

*snow-free season lengths explained 45 % of the variation in the trends in fire season length) (Fig. 4d)."*

5.  I could not understand what is written in 3.4 the Ignition timing and fire size, or what is expressed in Figure 5. It is necessary to explain the view shown in Fig. 5 in more detail in the text.

    **We understand that the section may not have conveyed the message clearly. We will restructure and rewrite this section for clarity in the revised manuscript.**

    *"Fire ignitions that occurred in the early fire season ($20^{th}$ percentile earliest ignitions) resulted in larger fires compared to fires that were ignited later in the season ($80^{th}$ percentile latest ignitions) in all ecoregions but the Alaska Tundra (Fig. 5B) and the Eastern Softwood Shield (Fig. 5P). This difference was significant in 8 out of the 16 ecoregions at $p < 0.1$ with the early ignited fires resulting in 77 % larger fires compared to fires ignited later in the season across the study domain (Fig. 5). On an ecoregional level, the early ignited fires grew between 30 and 600 % larger than late season fires. The largest difference in fire size between early ignited and late ignited fires were observed in the southern ecoregions (Table S9). Also, in these ecoregions, early-season fires accounted for more than half of the total burned area (Fig. 5 J, L, O and Table S9) whereas in the northern ecoregions early-season fires accounted for approximately one third of the total burned area. Across our study domain, the $20^{th}$ percentile earliest ignited fires accounted for an average of 40.6 (standard deviation: 14.2) % of the total annual burned area (Table S9). Nonetheless, the largest early ignited fires on average were observed in the forested ecoregions of Alaska Boreal Interior (Fig. 5A), Taiga Plain (Fig. 5E), and Western Taiga Shield (Fig. 5G) (23 218 (standard deviation: 7 557) ha) compared to the other ecoregions (9 922 (standard deviation: 5 192) ha))."*

    **Also, we will add extra information in the caption of figure 5 in the revised manuscript:**

    *"Figure 5 Fire size as a function of ignition timing for all ecoregions (A-P). The $20^{th}$ percentile day of ignition was set as threshold to discriminate between early (colored) and late season fires (gray). The colored dashed lines indicate the mean ignition timing and fire size for all early season ignitions while the gray dashed lines indicate the mean ignition timing and fire size for all late season ignitions. Significant difference between early and late ignited fires were indicated by * ($p < 0.1$) and ** ($p < 0.05$). Note the logarithmic scale for fire size."*

Comments on specific points are provided below.

Line 35-36: After "... are heterogeneous across the Northern Hemisphere" in this sentence, we suggest citing Suzuki et al. (2020, doi:10.1002/hyp.13844).

**We agree that the sentence needs references. We have added the Suzuki et al. 2020 and the Bormann et al., 2018 citations to show the regional differences across both Siberia and North America in line 37.**

*Suzuki, K., Hiyama, T., Matsuo, K., Ichii, K., Iijima, Y., & Yamazaki, D. (2020). Accelerated continental-scale snowmelt and ecohydrological impacts in the four largest Siberian river basins in response to spring warming. Hydrological Processes, 34(19), 3867-3881.*

*Bormann, K. J., Brown, R. D., Derksen, C., & Painter, T. H. (2018). Estimating snow-cover trends from space. Nature Climate Change, 8(11), 924-928.*

Line 73: We recommend adding Bartsch et al. (2009, doi:10.1088/1748-9326/4/4/045021) to this citation.

**The paper is a great addition to the sentence. We have added the reference to the revised manuscript.**

Line 108-140: The entire section needs to be revised because snow disappearance timing, and not snowmelt timing, is mentioned here.

**Thank you, we will revise the terminology manuscript and we will use the term snow disappearance timing in the revised manuscript.**

Line 220-223: To demonstrate the validity of using ERA5 surface data, please cite the application papers on ERA5 surface data used in such analyses.

**We agree and have will some text addressing this with adequate citations in the revision:**

> *"ERA5 reanalysis data have been used before in other studies that investigated extreme weather events and fires in the northern high latitudes (Gloege et al., 2022; Parisien et al., 2023). Furthermore, several of the ERA5 variables, such as precipitation, surface temperature, and specific humidity have been validated with ground observations over the study region (Alves et al., 2020)."*

Line 260: "andthe…" should be revised to "and the.."

**Thank you for noticing this. We will change it in the revision.**

Line 399-401: P may represent a narrow temporal window. Isn't it?

**Yes, that could indeed be categorized as having a narrower temporal window comparable to Fig 3 A-I, K. We will change this sentence in the revised manuscript to:**

*"Ignitions occurred later and in a narrower temporal window in the northern ecoregions (Fig. 3 A-I, K) and Eastern Softwood Shield (Fig. 3 P) compared to the other southern ecoregions. Southern ecoregions also showed a more variable ignition timing at the beginning of the fire season (Fig. 3 J, L-P)"*

Line 435-437: It does not appear consistent with Figure 4d.

**See answer to point 4 of the major comments.**

Line 447: "Boreal Interior, Taiga Plain, and Western Taiga Shield..." should be preferably revised to "Boreal Interior (Fig. 5A), Taiga Plain (Fig. 5E), and Western Taiga Shield (Fig. 5N)...".

**Thank you. We agree for the readability this is better and will change it accordingly in the revised manuscript.**

---

## Author Comment (AC2)

**We thank Dr. Quinn Barber for the thorough review of our paper. We have included the suggestions to improve the quality of the paper. The original comments are copied here with our responses in bold underneath.**

General comments

This study by Hessilt et al., titled "Geographically divergent trends in snowmelt timing and fire ignitions across boreal North America", investigated the temporal trend in snowmelt timing and fire season length across 16 ecoregions of North America, including Canada and Alaska. Using an ambitious combination of climatological data, fire polygons, MODIS snow-cover products, and MODIS active-fire detections, the authors discuss spatially-variable trends in snowmelt timing and the fire season. Using piecewise structural equation modelling, they show that the strongest factor driving snowmelt date is air temperature, which is in turn the strongest factor driving ignition timing, but that snowmelt timing also drives a cascade of effects including fuel desiccation and favourable fire weather, thereby further influencing ignition timing.

Overall, this manuscript is well-written and deserves publication in EGUsphere with minor changes. The authors have done an excellent job of presenting their research with good-quality writing and excellent figures. One general criticism I can offer is that one of the findings, a spatially-divergent trend in ignition timing, has been reported on an Ecozone-level in Hanes et al. (2019)(see Table 3), which has not been adequately cited and discussed here. There are also several methodological limitations which will need to be addressed before publication. In spite of these limitations, this manuscript advances understanding of drivers and trends in ignition and snowmelt seasonality, and is well worthy of publication in EGUsphere.

**Thank you for suggesting the Hanes et al. 2019 paper. We have included it in the introduction as a key reference to changes in area burned.**

**Line 44-45:** ***"Simultaneously, over the last two decades, large parts of western boreal North America have experienced a rise in the number of lightning fire ignitions and burned area (Hanes et al., 2019), driven by increases in (…)."***

**Line 54-56:** ***"The relationships between snow disappearance timing and fire behavior characteristics, such as fire ignitions and size, may vary across boreal North America and remain poorly understood (Hanes et al., 2019)."***

**Also, we included it in the discussion at multiple places:**

**Lines 528-529:** ***"The divergent trend in number of ignitions is in accordance with a previous study on changes in the number of fires and burned area across Canada from 1959 to 2015 (Hanes et al., 2019)."***

**Line 613-616:** ***"Other studies have, with the usage of governmental fire perimeter data, also found a prolonging of the fire season length limited to the western North America (Westerling et al., 2006, Albert-Green et al., 2014, Hanes et al., 2019)"***

Specific comments:

1. (Lines 115-117) The use of a 15% threshold for 14 consecutive days is problematic at high latitudes. As MODIS angle of observation increases, the degree of noise present in the signal also increases. I've attached a snapshot of Terra NDSI from 2018 at a random point in the Northwest

Territories, 65 degree latitude. It is apparent that the signal for defining snowmelt jumps above 15% NDSI even deep into August, which would confound the 14-consecutive day algorithm. One would expect this would cause erroneously-late fire snowmelt dates in the northern parts of the study area. However, since this error is applied equally across the study period 2001-2019, it is not likely to significantly bias the findings. I recognize that recalculating NDSI snowmelt timings is outside the scope of this paper, since the authors are using the Verbyla (2017) product. Therefore, I would recommend that the authors discuss the limitations of using MODIS-derived snowmelt products at high latitude in the discussion, possibly in a new "Limitations" section or integrated in the discussion.

**Thank you for considering this aspect. We agree that the retrieval of snow disappearance at high latitudes is influenced by the observation angle of MODIS. We have compared the snow disappearance timing from our initial dataset with a 14-day threshold to a snow disappearance timing from MODIS data with a 7-days threshold. When we compare the annual average snow disappearance timing per ignition and ecoregion, we did not find large differences between the two approaches to retrieve snow disappearance timing retrieval thresholds (Fig. S1). Moreover, the ignition disappearance timings retrieved using these two approaches strongly correlated (ignition: Pearson R = 0.98, ecoregion: Pearson R = 0.76) with generally a somewhat later snow disappearance timing using the 14-day per ecoregion (Fig. S5). Given that two retrieval approaches were strongly correlated, the effect of threshold choice on subsequent analysis steps (e.g. correlation with ignition timing) is expected to be small. Further, the largest discrepancy between snow disappearance timing is found in the southwestern ecoregions, contrary to the idea of larger signal noise in northern ecoregions (Table S1). This shows that the threshold of 14 days may be more robust to detect the actual snow disappearance timing. In ecoregions with variable snowfall, a threshold of fewer days may result in detecting early snow disappearance timing, which is then followed by new snow fall a few days later. In summary, we found that the threshold choice results in similar temporal patterns of snowmelt timing and that a longer threshold time may be more robust for the southern ecoregions. After this analysis, we decided to keep our original methodological choices in the revised manuscript.**

[Figure]

[Figure]

**Fig. S5** The annual average snow disappearance timing per ignition (A) and per (B) ecoregion for a retrieval threshold in which pixels need to be snow-free for seven vs. 14 days. The red line indicates the 1:1 line.

**Table 1** The mean difference per ecoregion and ignition for comparison of snow disappearance timing retrieval with a threshold of 7 days snow free pixel and to 14 days snow free pixel.

| Ecoregion | Ecoregion, mean difference | Ignition, mean difference |
|---|---|---|
| Alaska Boreal Interior | 3.96 | 0.04 |
| Alaska Tundra | 28.87 | 0.66 |
| Boreal Cordillera | 2.17 | 0.28 |
| Boreal Plain | 7.34 | -0.01 |
| Brooks Range Tundra | -0.05 | 0.41 |
| Cold Deserts | 23.23 | -0.22 |
| Hudson Plain | 0.16 | 0.07 |
| Marine West Coast Forest | 49.97 | 1.15 |
| Mixed Wood Shield | 10.08 | 0.12 |
| Eastern Softwood Shield | 14.75 | 0.08 |
| Western Softwood Shield | -0.39 | 0.02 |
| Taiga Cordillera | 1.86 | 0.05 |
| Taiga Plain | 5.20 | -0.05 |
| Eastern Taiga Shield | -0.15 | -0.01 |
| Western Taiga Shield | 1.42 | 0.09 |
| Western Cordillera | 26.82 | 0.13 |

**In the revision, we will also include a limitations section in the discussion, in which we will discuss how the selection of temporal threshold may influence the retrieval of snow disappearance timing.**

*"4.5 Limitations*
*We used a conservative threshold of 14 consecutive days of snow free pixels (NDSI ≤ 15 %) to calculate the snow disappearance timing. This could potentially influence the timing of snow disappearance to occur later than observed. A comparison of snow disappearance timing retrieval with a threshold of seven consecutive days of snow free pixels indicate that the retrievals resulted in similar temporal patterns in snow disappearance timing regardless of threshold choice (Fig. S11), with the 14-days threshold generally resulting in later snow disappearance timing. The largest discrepancies between the retrievals of snow disappearance timing with different temporal thresholds were found in the southern ecoregions (Table S13). This indicates that the threshold of 14 consecutive days with snow free pixels may be more*

*robust to determine snow disappearance timing, because of sudden changes in weather that causes snow offset and onset, especially in southern ecoregions."*

2. The use of the Canadian Large Fire Database (CLFD) is surprising, and very well may be a typo. The CLFD is a deprecated product which only includes fires from 1959 to 1999. The authors may be referencing the Canadian National Fire Database (CNFDB) or the National Burned Area Composite (NBAC), both available at https://cwfis.cfs.nrcan.gc.ca/datamart.

   **Indeed, we used the CNFDB database and not the deprecated CLFD version. We will change this in the revised manuscript.**

3. (Lines 169-171) Eliminating fires which occurred prior to snowmelt is a potentially confounding factor, and I would suggest this is the biggest weakness of the manuscript. Although fires occurring in deep winter are rightfully excluded, noise in the snowmelt data (see (1)) and long-lasting snow pockets can result in MODIS-detected snowmelt being delayed far after true fire season start (Pickell et al. 2017 provide discussion of this). That is to say, snowmelt detections have a high false negative rate, and should not be used to truncate ignitions except for those ignitions which are obviously impossible. The authors' calculation of correlation between fire season start and snowmelt is dependent on this assumption, since the authors use 1% cumulative fire ignitions to indicate fire season start. Impacts on trends in fire season length, and fire season start are also possible. I suggest that the authors rerun their calculations including only ignitions which occur at most 7, 14, or 21 days prior to snowmelt, selecting a threshold appropriately.

   **Thank you very much for this suggestion. The fraction of ignitions that occurred before snow disappearance and that we as such have excluded in our initial analysis accounted for between 0.1 to 6 % of the ignitions for the different ecoregions. Including ignitions that occurred between 7 and 21 days before snow disappearance slightly changes these numbers, but has negligible effects on the results.**

   **We also looked into how the inclusion of ignition points that occurred between 7 to 21 days before the snow disappearance timing would influence the 1$^{st}$ percentile day of ignition threshold we set for the fire season start. Specifically, we included all ignition points that occurred 7, 14, and 21 days before the snow disappearance timing occurred. We then calculated the 1$^{st}$ percentile day of the ignition, for each of these three scenarios, as set in our initial analysis of changes in the fire season length. We did not find substantial changes in the threshold set as fire season start when we included ignitions that occurred 7, 14 or 21 days before the snowmelt disappearance.**
   **The largest changes in the fire season threshold were observed when we used the 21 days threshold. These changes mainly occurred in the southern ecoregions that would be less prone to detection errors as mentioned in the referee's first major point, and more prone to human activity. Further, these regions already have fire season duration that exceeds the snow-free season and with inclusion of these ignitions prior to the snow disappearance timing we would add additional influence from these small anthropogenic fires.**

4. (Lines 169-179) Ignition causes are not in line with established proportions, as listed in the NBAC or NFDB. Not counting Alaska, approximately 17% of the fires from 1980-2019 over 200 ha in final area are classified as anthropogenic in the NFDB, although the authors report only 4% have an anthropogenic cause. It's possible that the authors are identifying some fires in ABoVE which are not represented in the NFDB, but it is very unlikely that the NFDB missed thousands of

fires over 200 ha in final area. This is difficult to disentangle without knowing what the true source of Canadian fire polygon data is, see (2).

**We had another look at this, but found similar results as we reported in the manuscript. The reason is that ABoVE-FED reports many small fires that were not reported in the CNFDB. We did not attribute an ignition cause to these fires. So, while the absolute number of anthropogenic and lightning fires in our study is comparable to the CNFDB, our study also includes a large number of small fires without cause attribution.**

5. (Lines 535 - 537) The idea that earlier snowmelt could lead to a persistent ridge is interesting, but is not supported by your results. This speculation should be cut, or discussion should be changed to indicate that it is speculation, with language such as "Although our results do not clearly indicate…"

**Thanks for this suggestion. We will change this accordingly to indicate that these ideas are based on speculations. We will add the following sentence in the revised manuscript:**

**"*Although our results do not provide clear indications, the persistent ridge formation could possibly also be a result of the divergent snow disappearance trend caused by the SSW.*"**

6. Although this manuscript is on "boreal North America", it includes the Western Cordillera which is not boreal for most of its area. The authors must change the text throughout to reflect that this study does not encompass only the boreal forest, such as using the word "super-boreal". Alternatively, a sentence in section 2.1 could be written to define the study area, such as 'hereafter, "boreal North America" for readability'.

**We will acknowledge in the methods section that the study region includes temperate ecosystems, but we will keep the term 'boreal North America' for readability. We have included a sentence saying:**

**"*Our study domain thus included Arctic tundra, boreal forest, and temperate ecosystems between Northwest Alaska and Southeast Canada, hereafter referred to as "boreal North America.*""**

7. Some further discussion of the study limitations is warranted, possible in a new 'Limitations' subsection but this is up to the authors discretion.

   1. The lack of significance in many Ecoregions is glossed over, and it should be mentioned that conclusions are being drawn from these insignificant trends (Fig 2). However it must be acknowledged that these trends are logical and are limited by the time series length, so are still worth reporting.

   2. The authors used pSEM well, but they provide insufficient evidence to conclude that early snowmelt is driving meteorological processes given the limited relationship strength for these variables and the interlinked nature of these variables. It is likely that this relationship varies between ecoregions, which are pooled in the pSEM. Some discussion of this limitation is needed.

   3. See item (1), the use of a 15% threshold for defining snowmelt.

We appreciate these interesting thoughts on the study's limitations. Regarding the point 7.2, we agree that we cannot conclude that the snow disappearance timing is directly driving weather variability. In the discussion, we had already acknowledged that the relationship is not strong in our study but indicates that there may be a connection, as reported in other studies (Gloege et al., 2022 and Scholten et al., 2022). Also, in the pSEM, we have used a linear-mixed-effect model with ecoregions as random effect. This accounts for the variations between ecoregions. We have moderated the language in the section 4.4, e.g. (lines 642-644):

"*Our model also suggests a cascading effect of snow disappearance timing on meteorological conditions that carried over into the influence on ignition timing.*"

We will include a discussion section addressing the study's limitations in the revised manuscript:

"*4.5 Limitations*
*We used a conservative threshold of 14 consecutive days of snow free pixels (NDSI ≤ 15 %) to calculate the snow disappearance timing. This could potentially influence the timing of snow disappearance to occur later than observed. A comparison of snow disappearance timing retrieval with a threshold of seven consecutive days of snow free pixels indicate that the retrievals resulted in similar temporal patterns in snow disappearance timing regardless of threshold choice (Fig. S11), with the 14-days threshold generally resulting in later snow disappearance timing. The largest discrepancies between the retrievals of snow disappearance timing with different temporal thresholds were found in the southern ecoregions (Table S13). This indicates that the threshold of 14 consecutive days with snow free pixels may be more robust to determine snow disappearance timing, because of sudden changes in weather that causes snow offset and onset, especially in southern ecoregions.*

*The long-term and short-term trends of snow disappearance timing and number of ignition were not consistently statistically significant for all ecoregions. The uncertainty related to the retrieval of both snow disappearance timing and fire perimeters from the pre-MODIS era (before 2000) resulted in large variation in both variables. More robust findings could potentially be drawn with longer time series. Also continued monitoring of snow disappearance and ignition timing is needed to track the relationship between these two variables as their relationship may become more pronounced with further climate change. Similarly, longer and more consistent time series could increase the robustness of the analysis on snow-free vs. fire season length. While we observed a significant relationship between these variables across ecoregions (Fig. 4c), this was not evident for most ecoregions (snow-free periods: p < 0.1 in 6 ecoregions and fire season length: p < 0.1 for 2 ecoregions). We only used the shorter time period between2001 and 2019 data to establish these changes as these represent the higher quality data during the MODIS era.*

*Lastly, we modelled complex land-atmosphere interactions, including relationships between snow disappearance timing, weather and ignition timing, using simple pSEMs. The interactions may be more complex and include other variables on different scales that we did not include in our analysis. Our models provide further support of the importance of land-atmosphere dynamics in relation to fire, yet our analysis did not provide robust relationships explaining the mechanistic interactions. We therefore call for further investigation of the interactions between snow disappearance timing, atmospheric conditions, and their influences on the fire season start, length, extent and severity.*"

**Technical comments:**

- (Lines 10-11) There is no consensus that fire severity has increased (Guindon et al. 2021), rephrase.

  **Thank you. We agree that indeed from the 1980s there no evidence that fire severity has increased across Canada. We will rephrase the sentence on lines 10-11 to:**

  **"*The snow cover extent across the Northern Hemisphere has diminished while the number of lightning strikes and area burned has increased over the last five decades with accelerated warming.*"**

- (Lines 18-19) "Earlier snowmelt induced earlier ignitions…" This topline result may be unduly influenced by the decision to cut all ignitions prior to snowmelt. Given the importance of this result it is necessary to recalculate including ignitions prior to MODIS snowmelt.

  **Thank you. We have addressed this matter in your specific comment point 3.**

- (Line 26) "The number of fires in eastern boreal North America" I'm not sure your results support this (Fig 2), it may be better to leave it out.

  **Thanks, this is a good point that surely could be investigated with more depth in another paper. We will rephrase the sentence on line 25 and 28 to:**

  **"*Future warming and consequent changes in snow disappearance timing may contribute to further increases in western boreal fires while it remains unclear how the number and timing of fire ignitions in eastern boreal North America may change with climate change.*".**

- (Line 260) Missing space between "and" and "the".

  **Thank you for noticing this. We will change it in the revision.**

- (Lines 352-357) These sentences need to be proofread, in particular the "promoted to a distinct" segment is very difficult to read and understand.

  **We agree that the message may not have been clearly conveyed. We will rephrase this section, thereby also including additional references as suggested elsewhere in this review:**

  **"*The long-term (1980-2019) and short-term (2001-2019) snow disappearance timing trends over boreal North America showed somewhat similar patterns. Long-term snow disappearance timing trends demonstrated shifts towards earlier snow disappearance timing in 13 of 16 ecoregions, but this trend was only significant in three ecoregions (p < 0.05) (Fig. 2a). These significant trends towards earlier snow disappearance were observed in Northwestern boreal North America ecoregions (Fig. 2b A-D), while three southern ecoregions (Boreal Plain, Mixed Wood Shield, and Eastern Softwood Shield) showed later snow disappearance timing between 1980 and 2019 (Fig. 2a M, O, P). Between 2001 and 2019, this spatial divergence in the snow disappearance timing trend has developed into a distinct west-east divergence across boreal North America, with increasingly earlier snow disappearance observed in western boreal North America versus later snow disappearance in the eastern ecoregions with only four ecoregions showing statistically significant changes (p < 0.05) between 2001 and 2019 (Figs. 1a and 2, and Table S1).*"**

- (Lines 459-461) You haven't shown that the subset pSEMs (Tables S11, S12) are weaker than the full pSEM (Table S10). Is statement based on AIC? AIC is sensitive to sample size, and so cannot be compared between the three models. It would be best to compare model fit using a comparative fit index such as one described in Hu & Bentler (1999). You could alternately cut this comparison entirely, as the intercomparison of the models is not a key finding.

  **We agree that such a model comparison would be sensitive to sample size and that this is not a main finding of our study. We will rephrase this section accordingly. In addition, we will clarify that the model for ignition timing was split between fire weather and weather variables, as suggested in one of the other reviewer comments.**

  "*The model fits for ecoregions with earlier snow disappearance timing trend (Fisher's $C_{86}$ = 96.31, p = 0.21) and later snow disappearance timing trends (Fisher's $C_{112}$ = 107.14, p = 0.61) showed similar patterns as the pSEM fit for all ecoregions (Fig. S7). The variance explained in the snow disappearance timing and ignition timing were generally better when splitting ecoregions between those with earlier and later snow disappearance trends. The pSEM model for earlier snow disappearance trends explained 32 % of the variation (M-$R^2$ = 0.32, C-$R^2$ = 0.32) while 54 % of the variation in ignition timing was explained by the model (fire weather: M-$R^2$ = 0.15, C-$R^2$ = 0.15, weather: M-$R^2$ = 0.39, C-$R^2$ = 0.39). The pSEM model for ecoregions with later snow disappearance trends explained 53 % of the variation in the snow disappearance timing (M-$R^2$ = 0.53, C-$R^2$ = 0.53) and 53 % of the variation in ignition timing (fire weather: M-$R^2$ = 0.18, C-$R^2$ = 0.18, weather: M-$R^2$ = 0.35, C-$R^2$ = 0.37) (Fig. S7).*"

- (Lines 476-478) This sentence is unclear. "DC influenced ignition timing positively" would suggest that higher DC delayed ignitions. "earlier ignitions generally occurred under wetter DC conditions" is also unclear, in the context of the pSEM. Please clarify.

  **You are correct in your observations. While it may be counterintuitive, DC has a very slow response time (around 52 days) to changes in weather and therefore builds up progressively over the fire season, see figure taken from the National Wildfire Coordinating Group (NWCG). Hence, earlier ignitions on average will have a lower DC than late ignitions.**

[Figure]

However, we have accommodated this request by clarifying the sentence:

*"For ecoregions with earlier snow disappearance timing, DC influenced the ignition timing positively meaning that earlier ignitions generally occurred when DC was still low."*

**and added an extra sentence in the discussion of the results:**

"*The fine fuel moisture and duff moisture codes showed significant influences on ignition timing, while the drought code did not, possibly due to its slower response to changes in weather (hypothesis 5).*"

- (Lines 481-482) You can abbreviate CAPE here.

   **Thank you for noticing this. We will change it in the revised manuscript.**

- (Lines 535-537) Fix this grammatical error, there are two sentences here that need to be split. Also, these processes describe the extensive eastern Canadian fires in 2023, and the authors may want to mention that, although it's not strictly necessary.

   **We will split up the sentence into two separate sentences in the revised manuscript. Also, we appreciate the comment on potentially adding a sentence on the recent 2023 fires in Canada:**

   "*These processes may also have triggered and influenced the persistency of the severe fire season across Canada in 2023.*"

- (Lines 617-622) These discussions neglect to mention that the southern and eastern ecoregions have a much higher deciduous proportion than the northwestern ecoregions. Although this is outside of the scope of the study, it's necessary to discuss that these are a factor influencing the drivers of early-season flammability, particularly in the Boreal Plains.

   **We will include this in our discussion for the final revised paper as this is an important factor for ignition efficiency.**

   *"The ecoregions with later snow disappearance timing, which showed less carry-over effect of snow disappearance timing on weather and fuel moisture to ignition, also corresponded to the more densely populated regions with larger fraction of deciduous trees (Pavlic et al., 2007; Olthof et al., 2015). The lack of carry-over effect may be due other drivers such as elevated potential for anthropogenic ignitions and the spring window during which deciduous forests are most flammable (Wotton et al., 2010; Parisien et al., 2023) ."*

- (Figure 1) In panel B, you have used the Ecoregions without a split in the softwood shield, this should be changed for consistency.

   **Thank you for noticing this. We will add the split in the Softwood Shield in the revised manuscript.**

- (Figure 3) The caption says that the ignitions on these plots correspond to the annual 20th percentile of ignitions, but some of the ecoRegions have ignitions extending past the 300th julian day. How is this possible? Is the caption wrong, or have you accidentally included all the ignitions in some of the plots? Please correct.

We had a deeper look into this. The 20$^{th}$ percentile was calculated annually. Some years may have very few ignitions, which may result in the 20$^{th}$ percentile occurring late in the fire season. This holds true for Western Cordillera and Cold Deserts, hence, the large variation in the day of ignition. We also mainly observe this pattern in the southern ecoregions indicating that some of these small fires are due to anthropogenic activity.

- (Figure 4) You are missing an "M" label in panel B. Panel D is not readable, please increase the size of the points significantly or change the points to their corresponding letter label (e.g. "A", "B").

  **We appreciate the suggestion. We will increase the size of the points in Panel D of a revised figure 4.**

- (Figure 5) The grey dashed line is too light and difficult to read. All dashed lines should be darkened and plotted on top of the data. Also, I'm assuming the coloured fire data represents early-season ignitions, while the grey data represents all other ignitions. If this is true, why do some coloured lines appear mixed into the late-season data (e.g. in panel L)? Please fix or clarify.

  **Thanks for the comment. We will revise the figure accordingly. Indeed, the colored bars represent early ignitions and the grey bars represent the late ignitions. We will make clear in the revised figure caption what the different color schemes represent. Also, the 20$^{th}$ percentile ignition is set annually as it may differ given different weather conditions or snow conditions which causes the bars to mix with later ignition days. We will clarify this in the methods on lines 210 to 212.**

  *"Thus, we set the ignition timing threshold to the annual 20$^{th}$ percentile of the ignition timing distribution to account for potential interannual differences in weather and snow disappearance timing interfering with the ignition timing."*

  And in the figure caption for figure 5:

  *"The 20$^{th}$ percentile day of ignition was set as threshold to discriminate between early (colored) and late season fires (gray)."*

- (Figure 6) Why are you reporting M-R2 and C-R2 separately for fire weather and for weather? Were these two models run independently? If so, please indicate it somewhere.

  **Indeed, since fire weather variables are derived from weather variables, we decided to run the models separately. We agree this was not clear from line 337.**

  **"*As the pSEMs can consist of many different linear models, we fitted each component of the pSEM with a linear mixed-effect model.*"**

  **We will include two sentences to clarify this:**

  **"*Therefore, the influence of fire weather and weather on ignition timing were modelled separately. The influence of snow disappearance timing was included in the model that contained weather variables predicting ignition timing.*"**

  **We will also clarify this in the figure caption:**

"*The influence of fire weather and weather on ignition timing were modelled separately.*"

- (Works cited) Most of your academic citations do not have a publication year, please add it.

  **We have looked at the references and could not find any references without a publication year. The perception of lack of publication years may stem from the journal's reference style, which places the publication year in the end, see for example, year in bold:**

  "*Abatzoglou, J. T. and Williams, A. P.: Impact of anthropogenic climate change on wildfire across western US forests, Proc. Natl. Acad. Sci. U. S. A., 113, 11770–11775, https://doi.org/10.1073/pnas.1607171113, **2016***".

**Works cited:**

Guindon, L., Gauthier, S., Manka, F., Parisien, M.A., Whitman, E., Bernier, P., Beaudoin, A., Villemaire, P. and Skakun, R., 2021. Trends in wildfire burn severity across Canada, 1985 to 2015. *Canadian Journal of Forest Research*, *51*(9), pp.1230-1244.

Hanes, C.C., Wang, X., Jain, P., Parisien, M.A., Little, J.M. and Flannigan, M.D., 2019. Fire-regime changes in Canada over the last half century. *Canadian Journal of Forest Research*, *49*(3), pp.256-269.

Hu, L. T., & Bentler, P. M. (1999). Cutoff criteria for fit indexes in covariance structure analysis: Conventional criteria versus new alternatives. *Structural equation modeling: a multidisciplinary journal*, *6*(1), 1-55.

Pickell, P.D., Coops, N.C., Ferster, C.J., Bater, C.W., Blouin, K.D., Flannigan, M.D. and Zhang, J., 2017. An early warning system to forecast the close of the spring burning window from satellite-observed greenness. *Scientific Reports*, *7*(1), p.14190.

Verbyla, D., 2017. ABoVE: Last Day of Spring Snow, Alaska, USA, and Yukon Territory, Canada, 2000-2016. *ORNL DAAC*.

---

## Author Response (AR1)

**We thank the reviewer for helpful and in-depth comments. We have addressed those in a revised version and believe that these changes make the manuscript stronger. In this file, the original comments are copied with our responses in bold underneath.**

Major Comments

The study by Zeng et al., titled "Geographically divergent trends in snowmelt 1 timing and fire ignitions across boreal North America," reported the influence of snowmelt timing on fire ignitions across the ecoregions of boreal North America. They found spatially divergent trends in early (late) snowmelt that led to an increasing (decreasing) number of ignitions in the northwestern (southeastern) ecoregions between 1980 and 2019. Early snowmelt is a proxy for early ignition but may also result in a cascade of effects from early desiccation of fuels and favorable weather conditions that lead to earlier ignition. This indicates that snowmelt timing is an important trigger for land–atmosphere dynamics.

Overall, this paper is logical and worthy of publication. However, minor revisions are required prior to publication.

Please note the following points.

1. The major limitation of this paper was the incorrect use of the term "snowmelt." While snowmelt is also associated with snow cover, the authors must understand that the majority of snowmelt occurs at 100% snow cover. The term "snowmelt" used in this paper is incorrect and should be corrected throughout the paper, as it is more accurate than the terms "snow disappearance date" or "snow disappearance timing." Otherwise, the readers may fail to understand the authors' analyses. I believe that the title also needs to be revised.

   **We appreciate the thorough explanation and we have now used the term snow disappearance in the revised manuscript.**

2. The authors used surface data from ERA5 as the climate drivers of snowmelt and ignition timing, but ERA5 is a model estimate, and surface data are known to have bias. The bias is particularly large at high latitudes because of the lack of land-based observations and thus differs from ground-based data. It is necessary to demonstrate the validity of using ERA5 in this study with a reference.

   **Thank you. We have added a paragraph in our method section demonstrating that the use of ERA5 reanalysis data is acceptable for high latitude regions. Lines 231-234:**

   *"ERA5 reanalysis data have been used before in other studies that investigated extreme weather events and fires in the northern high latitudes (Gloege et al., 2022; Parisien et al., 2023). Furthermore, several of the ERA5 variables, such as precipitation, surface temperature, and specific humidity have been validated with ground observations over the study region (Alves et al., 2020).*

   *Alves, M., Nadeau, D. F., Music, B., Anctil, F., & Parajuli, A. (2020). On the performance of the Canadian Land Surface Scheme driven by the ERA5 reanalysis over the Canadian boreal forest. Journal of Hydrometeorology, 21(6), 1383-1404.*

   *Gloege, L., Kornhuber, K., Skulovich, O., Pal, I., Zhou, S., Ciais, P., & Gentine, P. (2022). Land-Atmosphere Cascade Fueled the 2020 Siberian Heatwave. AGU Advances, 3(6), e2021AV000619.*

   *Parisien, M. A., Barber, Q. E., Flannigan, M. D., & Jain, P. (2023). Broadleaf tree phenology and springtime wildfire occurrence in boreal Canada. Global Change Biology.*

3. (4) Surface relative humidity was used to model snowmelt timing, but I do not consider it as a good indicator of atmospheric dryness because surface relative humidity varies significantly depending on temperature. Instead, I recommend using the surface saturation deficit. The model in Figure 6 should also be recalculated using the saturation deficit because temperature and relative humidity vary almost identically, which is not desirable as a variable in a hypothesized model.

**The surface saturation deficit is in essence very similar to vapor pressure deficit that is often used in fire studies. Both variables quantify the difference between the available moisture in the air and the air's total moisture capacity at saturation. In response to your comments, we have considered an alternative model in which we modelled snow disappearance timing with vapor pressure deficit (VPD) rather than relative humidity (RH).**

[Figure]

**Fig. 1S The piecewise structural equation model as the initial model except that relative humidity is substituted by vapor pressure deficit.**

**The overall model performance remained the same regardless of variable. The prediction of snow disappearance timing was similar using VPD or RH. The substitution of RH with VPD**

in prediction of ignition timing did not change the direction of convective available potential energy (CAPE), VPD, and temperature but slightly diminished their effect ($p > 0.05$) (Fig. 1S, not included in the revised manuscript). Only snow disappearance timing and precipitation remained significant contributors of earlier ignition timing ($p < 0.05$). Further, snow disappearance timing influenced VPD, similar as with RH. Part of this can be explained by the correlation between VPD and temperature as VPD was derived from RH and temperature. We obtained similar results when modelling snow disappearance and ignition timing for ecoregions with earlier and later snowmelt timing when using VPD instead of RH (Figure S7 original manuscript). The inclusion of VPD diminished the influence of other weather variables on ignition timing (Fig. 1S). The only difference was that VPD influenced ignition timing insignificantly ($p > 0.05$) compared to the model with RH in ecoregions with later snowmelt timing (Fig. S7b original manuscript). The diminished effect of other weather variables was caused by the effect of VPD.

Therefore, we also ran a model in which temperature was excluded as it is also indirectly represented by the VPD (Fig. S9 original manuscript). For all pSEMs, we obtained somewhat similar results as observed for the models including air temperature and relative humidity (Fig. S8 and S9).

[Figure]

**Fig. S9 The piecewise structural equation model for all ecoregions, and for ecoregions with earlier and later snow disappearance timing as the initial model but relative humidity and temperature are substituted by vapor pressure deficit.**

**In summary, we understand the reviewer's comment, but found that the inclusion of VPD did result in very similar model structure as the original models with RH, albeit with slightly lower performance. We therefore kept our models with RH as the main analysis in the revised manuscript, and we have added the alternative models with VPD as supplementary figure S9.**

4. Figure 4d shows that there was more variation than a positive correlation, but we cannot say that there was a positive correlation. I would like to have this figure removed or a rationale added to say that the correlation was strong for those with large variations.

   **We agree that there is large variation in the data. We kept the figure 4d but have included a paragraph describing the correlation and variation we observed in the data. Lines: 465-469:**

*"We also found that the trends in snow-free and fire season length tended to correlate positively with each other with a prolonging of the fire season of 0.9 days per decade for every day per decade increase in the snow-free season (p = 0.08). There was variation between ecoregions and the trends in snow-free season lengths explained 45 % of the variation in the trends in fire season length (Fig. 4d)."*

5. I could not understand what is written in 3.4 the Ignition timing and fire size, or what is expressed in Figure 5. It is necessary to explain the view shown in Fig. 5 in more detail in the text.

**We understand that the section may not have conveyed the message clearly. We have restructured and rewritten this section for clarity in the revised manuscript, lines (472-486).**

*"Fire ignitions that occurred in the early fire season ($20^{th}$ percentile earliest ignitions) resulted in larger fires compared to fires that were ignited later in the season ($80^{th}$ percentile latest ignitions) in all ecoregions but the Alaska Tundra (Fig. 5 B) and the Eastern Softwood Shield (Fig. 5 P). This difference was significant in 8 out of the 16 ecoregions at $p < 0.1$ with the early ignited fires resulting in 77 % larger fires compared to fires ignited later in the season across the study domain (Fig. 5). On an ecoregional level, the early ignited fires grew between 30 and 600 % larger than late season fires. The largest difference in fire size between early ignited and late ignited fires was observed in the southern ecoregions (Table S10). Also, in these ecoregions, early-season fires accounted for more than half of the total burned area (Fig. 5 J, L, O and Table S10) whereas in the northern ecoregions early-season fires accounted for approximately one third of the total burned area. Across our study domain, the $20^{th}$ percentile earliest ignited fires accounted for an average of 40.6 (standard deviation: 14.2) % of the total annual burned area (Table S10). Nonetheless, the largest early ignited fires on average were observed in the forested ecoregions of Alaska Boreal Interior (Fig. 5A), Taiga Plain (Fig. 5E), and Western Taiga Shield (Fig. 5G) (23 218 (standard deviation: 7 557) ha) compared to the other ecoregions (9 922 (standard deviation: 5 192) ha))"*

**Also, we added extra information in the caption of figure 5 in the revised manuscript, lines (766-770):**

*"Figure 5 Fire size as a function of ignition timing for all ecoregions (A-P). The $20^{th}$ percentile day of ignition was set as threshold to discriminate between early (colored) and late season fires (gray). The colored dashed lines indicate the mean ignition timing and fire size for all early season ignitions while the gray dashed lines indicate the mean ignition timing and fire size for all late season ignitions. Significant difference between early and late ignited fires were indicated by * ($p < 0.1$) and ** ($p < 0.05$). Note the logarithmic scale for fire size."*

Comments on specific points are provided below.

Line 35-36: After "... are heterogeneous across the Northern Hemisphere" in this sentence, we suggest citing Suzuki et al. (2020, doi:10.1002/hyp.13844).

**We agree that the sentence needs references. We have added the Suzuki et al. 2020 and the Bormann et al., 2018 citations to show the regional differences across both Siberia and North America in line 39.**

*Suzuki, K., Hiyama, T., Matsuo, K., Ichii, K., Iijima, Y., & Yamazaki, D. (2020). Accelerated continental-scale snowmelt and ecohydrological impacts in the four largest Siberian river basins in response to spring warming. Hydrological Processes, 34(19), 3867-3881.*

*Bormann, K. J., Brown, R. D., Derksen, C., & Painter, T. H. (2018). Estimating snow-cover trends from space. Nature Climate Change, 8(11), 924-928.*

Line 73: We recommend adding Bartsch et al. (2009, doi:10.1088/1748-9326/4/4/045021) to this citation.

**The paper is a great addition to the sentence. We have added the reference to the revised manuscript.**

Line 108-140: The entire section needs to be revised because snow disappearance timing, and not snowmelt timing, is mentioned here.

**Thank you, we revised the terminology manuscript and we have used the term snow disappearance timing in the revised manuscript.**

Line 220-223: To demonstrate the validity of using ERA5 surface data, please cite the application papers on ERA5 surface data used in such analyses.

**We agree and have added some text addressing this with adequate citations in the revision, lines 231-234:**

> *"ERA5 reanalysis data have been used before in other studies that investigated extreme weather events and fires in the northern high latitudes (Gloege et al., 2022; Parisien et al., 2023). Furthermore, several of the ERA5 variables, such as precipitation, surface temperature, and specific humidity have been validated with ground observations over the study region (Alves et al., 2020)."*

Line 260: "andthe…" should be revised to "and the.."

**Thank you for noticing this. We changed it in the revision.**

Line 399-401: P may represent a narrow temporal window. Isn't it?

**Yes, that could indeed be categorized as having a narrower temporal window comparable to Fig 3 A-I, K. We changed this sentence in the revised manuscript (lines 429-431) to:**

*Ignitions occurred later and in a narrower temporal window in the northern ecoregions (Fig. 3 A-I, K) and Eastern Softwood Shield (Fig. 3 P) compared to the other southern ecoregions. Southern ecoregions also showed a more variable ignition timing at the beginning of the fire season (Fig. 3 J, L-P)."*

Line 435-437: It does not appear consistent with Figure 4d.

**See answer to point 4 of the major comments.**

Line 447: "Boreal Interior, Taiga Plain, and Western Taiga Shield..." should be preferably revised to "Boreal Interior (Fig. 5A), Taiga Plain (Fig. 5E), and Western Taiga Shield (Fig. 5N)...".

**Thank you. We agree for the readability this is better and have changed it accordingly in the revised manuscript.**

**We thank Dr. Quinn Barber for the thorough review of our paper. We have included the suggestions to improve the quality of the paper. The original comments are copied here with our responses in bold underneath.**

General comments

This study by Hessilt et al., titled "Geographically divergent trends in snowmelt timing and fire ignitions across boreal North America", investigated the temporal trend in snowmelt timing and fire season length across 16 ecoregions of North America, including Canada and Alaska. Using an ambitious combination of climatological data, fire polygons, MODIS snow-cover products, and MODIS active-fire detections, the authors discuss spatially-variable trends in snowmelt timing and the fire season. Using piecewise structural equation modelling, they show that the strongest factor driving snowmelt date is air temperature, which is in turn the strongest factor driving ignition timing, but that snowmelt timing also drives a cascade of effects including fuel desiccation and favourable fire weather, thereby further influencing ignition timing.

Overall, this manuscript is well-written and deserves publication in EGUsphere with minor changes. The authors have done an excellent job of presenting their research with good-quality writing and excellent figures. One general criticism I can offer is that one of the findings, a spatially-divergent trend in ignition timing, has been reported on an Ecozone-level in Hanes et al. (2019)(see Table 3), which has not been adequately cited and discussed here. There are also several methodological limitations which will need to be addressed before publication. In spite of these limitations, this manuscript advances understanding of drivers and trends in ignition and snowmelt seasonality, and is well worthy of publication in EGUsphere.

**Thank you for suggesting the Hanes et al. 2019 paper. We have included it in the introduction as a key reference to changes in area burned.**

**Line 45-48: *"Simultaneously, over the last two decades, large parts of western boreal North America have experienced a rise in the number of lightning fire ignitions and burned area (Hanes et al., 2019), driven by increases in dry fuel availability (Abatzoglou et al., 2016; Hessilt et al., 2022), favorable fire weather (Sedano and Randerson, 2014), and increase in the number of lightning strikes (Veraverbeke et al., 2017)"***

**Line 55-57: *"The relationships between snow disappearance timing and fire behavior characteristics, such as fire ignitions and size, may vary across boreal North America and remain poorly understood (Hanes et al., 2019)."***

**Also, we included it in the discussion at multiple places:**

**Lines 539-540: *"The divergent trend in number of ignitions is in accordance with a previous study on changes in the number of fires and burned area across Canada from 1959 to 2015 (Hanes et al., 2019)."***

**Line 625-627: *"Other studies that examined governmental fire perimeter data also found a prolonging of the fire season length limited to the western North America (Westerling et al., 2006; Albert-Green et al., 2013; Hanes et al., 2019)"***

Specific comments:

1. (Lines 115-117) The use of a 15% threshold for 14 consecutive days is problematic at high latitudes. As MODIS angle of observation increases, the degree of noise present in the signal also increases. I've attached a snapshot of Terra NDSI from 2018 at a random point in the Northwest Territories, 65 degree latitude. It is apparent that the signal for defining snowmelt

jumps above 15% NDSI even deep into August, which would confound the 14-consecutive day algorithm. One would expect this would cause erroneously-late fire snowmelt dates in the northern parts of the study area. However, since this error is applied equally across the study period 2001-2019, it is not likely to significantly bias the findings. I recognize that recalculating NDSI snowmelt timings is outside the scope of this paper, since the authors are using the Verbyla (2017) product. Therefore, I would recommend that the authors discuss the limitations of using MODIS-derived snowmelt products at high latitude in the discussion, possibly in a new "Limitations" section or integrated in the discussion.

**Thank you for considering this aspect. We agree that the retrieval of snow disappearance at high latitudes is influenced by the observation angle of MODIS. We have compared the snow disappearance timing from our initial dataset with a 14-day threshold to a snow disappearance timing from MODIS data with a 7-days threshold. When we compared the annual average snow disappearance timing per ignition and ecoregion, we did not find large differences between the two approaches to retrieve snow disappearance timing retrieval thresholds (Fig. S1). Moreover, the ignition disappearance timings retrieved using these two approaches strongly correlated (ignition: Pearson R = 0.98, ecoregion: Pearson R = 0.76) with generally a somewhat later snow disappearance timing using the 14-day per ecoregion (Fig. S1). Given that two retrieval approaches were strongly correlated, the effect of threshold choice on subsequent analysis steps (e.g. correlation with ignition timing) was expected to be small. Further, the largest discrepancy between snow disappearance timing was found in the southwestern ecoregions, contrary to the idea of larger signal noise in northern ecoregions (Table S2 in revision). This showed that the threshold of 14 days may have been more robust to detect the actual snow disappearance timing. In ecoregions with variable snowfall, a threshold of fewer days may have resulted in detecting prematurely snow disappearance timing that was then followed by new snowfall a few days later. In summary, we found that the threshold choice results in similar temporal patterns of snowmelt timing and that a longer threshold time may be more robust for the southern ecoregions. After this analysis, we decided to keep our original methodological choices in the revised manuscript.**

[Figure]

**Figure S1** The comparison of snow disappearance timing retrievals between 2001 and 2019 from MODIS with a threshold of 7 days and 14 days snow free pixels. The mean difference between the seven consecutive days of snow disappearance subtracted from the 14 consecutive days of snow disappearance, the Pearson correlation coefficient, and its significance (*p*) are indicated.

**Table S2 in revision** The mean difference (seven consecutive days of snow disappearance subtracted from 14 consecutive days of snow disappearance) per ecoregion and ignition for comparison of snow disappearance timing retrieval with a threshold of seven days snow free pixel and to 14 days snow free pixel.

| Ecoregion | Ecoregion, mean difference | Ignition, mean difference |
|---|---|---|
| Alaska Boreal Interior | 3.96 | 0.04 |
| Alaska Tundra | 28,87 | 0.66 |
| Boreal Cordillera | 2.17 | 0.28 |
| Boreal Plain | 7.34 | -0.01 |
| Brooks Range Tundra | -0.05 | 0.41 |
| Cold Deserts | 23.23 | -0.22 |
| Hudson Plain | 0.16 | 0.07 |
| Marine West Coast Forest | 49.97 | 1.15 |
| Mixed Wood Shield | 10.08 | 0.12 |
| Eastern Softwood Shield | 14.75 | 0.08 |
| Western Softwood Shield | -0.39 | 0.02 |
| Taiga Cordillera | 1.86 | 0.05 |
| Taiga Plain | 5.20 | -0.05 |
| Eastern Taiga Shield | -0.15 | -0.01 |
| Western Taiga Shield | 1.42 | 0.09 |
| Western Cordillera | 26.82 | 0.13 |

**In the revision, we also included a limitations section in the discussion, in which we discussed how the selection of temporal threshold may have influenced the retrieval of snow disappearance timing. Lines 687-697:**

*"4.5 Limitations*
*We used a conservative threshold of 14 consecutive days of snow free pixels (NDSI ≤ 15 %) to calculate the snow disappearance timing. This could potentially influence the timing of snow disappearance to occur later than observed. A comparison of snow disappearance timing retrieval with a threshold of seven consecutive days of snow free pixels indicated that the retrievals resulted in similar temporal patterns in snow disappearance timing regardless of threshold choice (Fig. S1, Table S2), with the 14-days threshold generally resulting in later snow disappearance timing. The largest discrepancies between the retrievals of snow disappearance timing with different temporal thresholds were found in the southern ecoregions (Table S2). This indicates that the threshold of 14 consecutive days with snow free pixels may be more robust to determine snow disappearance timing, because of sudden changes in weather can manifest in snow offset and onset, especially in southern ecoregions."*

2. The use of the Canadian Large Fire Database (CLFD) is surprising, and very well may be a typo. The CLFD is a deprecated product which only includes fires from 1959 to 1999. The authors may be referencing the Canadian National Fire Database (CNFDB) or the National Burned Area Composite (NBAC), both available at https://cwfis.cfs.nrcan.gc.ca/datamart.

   **Indeed, we used the CNFDB database and not the deprecated CLFD version. We changed this in the revised manuscript.**

3. (Lines 169-171) Eliminating fires which occurred prior to snowmelt is a potentially confounding factor, and I would suggest this is the biggest weakness of the manuscript. Although fires occurring in deep winter are rightfully excluded, noise in the snowmelt data (see (1)) and long-lasting snow pockets can result in MODIS-detected snowmelt being delayed far after true fire season start (Pickell et al. 2017 provide discussion of this). That is to say, snowmelt detections have a high false negative rate, and should not be used to truncate ignitions except for those ignitions which are obviously impossible. The authors' calculation of correlation between fire season start and snowmelt is dependent on this assumption, since the authors use 1% cumulative fire ignitions to indicate fire season start. Impacts on trends in

fire season length, and fire season start are also possible. I suggest that the authors rerun their calculations including only ignitions which occur at most 7, 14, or 21 days prior to snowmelt, selecting a threshold appropriately.

**Thank you very much for this suggestion. The fraction of ignitions that occurred before snow disappearance and that we as such have excluded in our initial analysis accounted for between 0.1 to 6 % of the ignitions for the different ecoregions. Including ignitions that occurred between 7 and 21 days before snow disappearance slightly changed these numbers, but had negligible effects on the results.**

**We also looked into how the inclusion of ignition points that occurred between 7 to 21 days before the snow disappearance timing would influence the 1st percentile day of ignition threshold we set for the fire season start. Specifically, we included all ignition points that occurred 7, 14, and 21 days before the snow disappearance timing occurred. We then calculated the 1st percentile day of the ignition, for each of these three scenarios, as set in our initial analysis of changes in the fire season length. We did not find substantial changes in the threshold set as fire season start when we included ignitions that occurred 7, 14 or 21 days before the snowmelt disappearance.**
**The largest changes in the fire season threshold were observed when we used the 21 days threshold. These changes mainly occurred in the southern ecoregions that would be less prone to detection errors as mentioned in the referee's first major point, and more prone to human activity. Further, these regions already have fire season duration that exceeds the snow-free season and with inclusion of these ignitions prior to the snow disappearance timing we would add additional influence from these small anthropogenic fires.**

4.  (Lines 169-179) Ignition causes are not in line with established proportions, as listed in the NBAC or NFDB. Not counting Alaska, approximately 17% of the fires from 1980-2019 over 200 ha in final area are classified as anthropogenic in the NFDB, although the authors report only 4% have an anthropogenic cause. It's possible that the authors are identifying some fires in ABoVE which are not represented in the NFDB, but it is very unlikely that the NFDB missed thousands of fires over 200 ha in final area. This is difficult to disentangle without knowing what the true source of Canadian fire polygon data is, see (2).

    **We had another look at this, but found similar results as we reported in the manuscript. The reason is that ABoVE-FED reports many small fires that were not reported in the CNFDB. We did not attribute an ignition cause to these fires. So, while the absolute number of anthropogenic and lightning fires in our study is comparable to the CNFDB, our study also includes a large number of small fires without cause attribution.**

5.  (Lines 535 - 537) The idea that earlier snowmelt could lead to a persistent ridge is interesting, but is not supported by your results. This speculation should be cut, or discussion should be changed to indicate that it is speculation, with language such as "Although our results do not clearly indicate…"

    **Thanks for this suggestion. We changed this accordingly to indicate that these ideas are based on speculations. We also added the following sentence in the revised manuscript, lines 581-583:**

    **"*Although our results do not provide clear indications, the persistent ridge formation could possibly also be a result of the divergent snow disappearance trend caused by the SSW.*"**

6.  Although this manuscript is on "boreal North America", it includes the Western Cordillera which is not boreal for most of its area. The authors must change the text throughout to reflect

that this study does not encompass only the boreal forest, such as using the word "super-boreal". Alternatively, a sentence in section 2.1 could be written to define the study area, such as 'hereafter, "boreal North America" for readability'.

**We acknowledged in the methods section that the study region includes temperate ecosystems, but we kept the term 'boreal North America' for readability. We have included a sentence (lines: 109-111) saying:**

*"Our study domain thus included Arctic tundra, boreal forest, and temperate ecosystems between Northwest Alaska and Southeast Canada, hereafter referred to as "boreal North America.""*

7.  Some further discussion of the study limitations is warranted, possible in a new 'Limitations' subsection but this is up to the authors discretion.

    1.  The lack of significance in many Ecoregions is glossed over, and it should be mentioned that conclusions are being drawn from these insignificant trends (Fig 2). However it must be acknowledged that these trends are logical and are limited by the time series length, so are still worth reporting.

    2.  The authors used pSEM well, but they provide insufficient evidence to conclude that early snowmelt is driving meteorological processes given the limited relationship strength for these variables and the interlinked nature of these variables. It is likely that this relationship varies between ecoregions, which are pooled in the pSEM. Some discussion of this limitation is needed.

    3.  See item (1), the use of a 15% threshold for defining snowmelt.

**We appreciate these interesting thoughts on the study's limitations. Regarding the point 7.2, we agree that we cannot conclude that the snow disappearance timing is directly driving weather variability. In the discussion, we had already acknowledged that the relationship is not strong in our study but indicated that there may be a connection, as reported in other studies (Gloege et al., 2022 and Scholten et al., 2022). Also, in the pSEM, we have used a linear-mixed-effect model with ecoregions as random effect. This accounted for the variations between ecoregions. We have moderated the language in the section 4.4, e.g. (lines 656-658):**

*"Our model also suggests a cascading effect of snow disappearance timing on meteorological conditions that carried over into the influence on ignition timing."*

**We included a discussion section addressing the study's limitations in the revised manuscript (lines: 687-722):**

*"4.5 Limitations*
*We used a conservative threshold of 14 consecutive days of snow free pixels (NDSI ≤ 15 %) to calculate the snow disappearance timing. This could potentially influence the timing of snow disappearance to occur later than observed. A comparison of snow disappearance timing retrieval with a threshold of seven consecutive days of snow free pixels indicates that the retrievals resulted in similar temporal patterns in snow disappearance timing regardless of threshold choice (Fig. S1, Table S2), with the 14-days threshold generally resulting in later snow disappearance timing. The largest discrepancies between the retrievals of snow disappearance timing with different temporal thresholds were found in the southern ecoregions (Table S2). This indicates that the threshold of 14 consecutive days with snow free pixels may be more robust to determine snow disappearance timing, because of sudden changes in weather can manifest in snow offset and onset, especially in southern ecoregions.*

*The long-term and short-term trends of snow disappearance timing and number of ignition were not consistently statistically significant for all ecoregions. The uncertainty related to the retrieval of both snow disappearance timing and fire perimeters from the pre-MODIS era (before 2000) resulted in large variation in both variables. More robust findings could potentially be drawn with longer time series. Also continued monitoring of snow disappearance and ignition timing is needed to track the relationship between these two variables as their relationship may become more pronounced with further climate change. Similarly, longer and more consistent time series could increase the robustness of the analysis on snow-free vs. fire season length. While we observed a significant relationship between these variables across ecoregions (Fig. 4c), this was not evident for most ecoregions (snow-free periods: p < 0.1 in six ecoregions and fire season length: p < 0.1 for two ecoregions). We only used the shorter time period between 2001 and 2019 data to establish these changes as these represent the higher quality data during the MODIS era.*

*Lastly, our pSEM analysis gives a simplified overview of relationships between snow disappearance timing, land-atmosphere dynamics, and fire ignitions. However, we acknowledge that these interactions are highly coupled. The complexity is beyond our model and may involve variables that we did not include. The influence of snow disappearance timing on atmospheric variables through surface albedo change and altered soil moisture may be difficult to decouple from the atmospheric variables and their persistent seasonal patterns on snow disappearance timing itself. Our models provide further support of the importance of land-atmosphere dynamics in relation to fire, yet our analysis did not provide robust relationships explaining the mechanistic interactions. We therefore call for a better understanding of the role of snow disappearance timing on land-atmospheric dynamics affecting boreal fires. Specifically, large-scale influence of continuous snow disappearance on soil and fuel properties, e.g. soil and fuel moisture, and atmospheric conditions e.g. vapor pressure deficit, and vice versa. Understanding these interactions and feedbacks could further advance our comprehension of how climate change is affecting changing boreal fire regimes."*

**Technical comments:**

- (Lines 10-11) There is no consensus that fire severity has increased (Guindon et al. 2021), rephrase.

  **Thank you. We agree that indeed from the 1980s there is no evidence that fire severity has increased across Canada. We rephrased the sentence on lines 10-11 to:**

  **"*The snow cover extent across the Northern Hemisphere has diminished while the number of lightning ignitions and burned area have increased over the last five decades with accelerated warming.*"**

- (Lines 18-19) "Earlier snowmelt induced earlier ignitions…" This topline result may be unduly influenced by the decision to cut all ignitions prior to snowmelt. Given the importance of this result it is necessary to recalculate including ignitions prior to MODIS snowmelt.

  **Thank you. We have addressed this matter in your specific comment point 3.**

- (Line 26) "The number of fires in eastern boreal North America" I'm not sure your results support this (Fig 2), it may be better to leave it out.

  **Thanks, this is a good point that surely could be investigated with more depth in another paper. We have rephrased the sentence on line 26 and 28 to:**

*"Future warming and consequent changes in snow disappearance timing may contribute to further increases in western boreal fires while it remains unclear how the number and timing of fire ignitions in eastern boreal North America may change with climate change.".*

- (Line 260) Missing space between "and" and "the".

  **Thank you for noticing this. We changed it in the revision.**

- (Lines 352-357) These sentences need to be proofread, in particular the "promoted to a distinct" segment is very difficult to read and understand.

  **We agree that the message may not have been clearly conveyed. We rephrased this section, thereby also included additional references as suggested elsewhere in this review, lines 375-386):**

  *"The long-term (1980-2019) and short-term (2001-2019) snow disappearance timing trends over boreal North America showed somewhat similar patterns. Long-term snow disappearance timing trends demonstrated shifts towards earlier snow disappearance timing in 13 out of 16 ecoregions, but this trend was only significant in three ecoregions (p < 0.05) (Fig. 2a). These significant trends towards earlier snow disappearance were observed in Northwestern boreal North America ecoregions (Fig. 2b A-D) while three southern ecoregions (Boreal Plain, Mixed Wood Shield, and Eastern Softwood Shield) showed later snow disappearance timing between 1980 and 2019 (Fig. 2a M, O, P). Between 2001 and 2019, this spatial divergence in the trends of snow disappearance timing has developed into a distinct west-east divergence across boreal North America. We observed increasingly earlier snow disappearance observed in western boreal North America versus later snow disappearance in the eastern ecoregions, with only four ecoregions showing statistically significant changes (p < 0.05) between 2001 and 2019 (Figs. 1a and 2, and Table S1)."*

- (Lines 459-461) You haven't shown that the subset pSEMs (Tables S11, S12) are weaker than the full pSEM (Table S10). Is statement based on AIC? AIC is sensitive to sample size, and so cannot be compared between the three models. It would be best to compare model fit using a comparative fit index such as one described in Hu & Bentler (1999). You could alternately cut this comparison entirely, as the intercomparison of the models is not a key finding.

  **We agree that such a model comparison would be sensitive to sample size and that this is not a main finding of our study. We rephrased this section accordingly. In addition, we clarified that the model for ignition timing was split between fire weather and weather variables, as suggested in one of the other reviewer comments, lines 492-502:**

  *"The model fits for ecoregions with earlier snow disappearance timing trend (Fisher's $C_{86}$ = 96.31, p = 0.21) and later snow disappearance timing trends (Fisher's $C_{112}$ = 107.14, p = 0.61) showed similar patterns as the pSEM fit for all ecoregions (Fig. S8). The variance explained in the snow disappearance timing and ignition timing were generally better when splitting ecoregions between those with earlier and later snow disappearance trends. The pSEM model for earlier snow disappearance trends explained 32 % of the variation (M-$R^2$ = 0.32, C-$R^2$ = 0.32) while 54 % of the variation in ignition timing was explained by the model (fire weather: M-$R^2$ = 0.15, C-$R^2$ = 0.15, weather: M-$R^2$ = 0.39, C-$R^2$ = 0.39). The pSEM model for ecoregions with later snow disappearance trends explained 53 % of the variation in the snow disappearance timing (M-$R^2$ = 0.53, C-$R^2$ = 0.53) and 53 % of the variation in ignition timing (fire weather: M-$R^2$ = 0.18, C-$R^2$ = 0.18, weather: M-$R^2$ = 0.35, C-$R^2$ = 0.37) (Fig. S8)."*

- (Lines 476-478) This sentence is unclear. "DC influenced ignition timing positively" would suggest that higher DC delayed ignitions. "earlier ignitions generally occurred under wetter DC conditions" is also unclear, in the context of the pSEM. Please clarify.

  **You are correct in your observations. While it may be counterintuitive, DC has a very slow response time (around 52 days) to changes in weather and therefore builds up progressively over the fire season, see figure taken from the National Wildfire Coordinating Group (NWCG). Hence, earlier ignitions on average have a lower DC than late ignitions.**

[Figure]

  **However, we have accommodated this request by clarifying the sentence, lines 516-518:**

  *"For ecoregions with earlier snow disappearance timing, DC influenced the ignition timing positively meaning that earlier ignitions generally occurred when DC was still low."*

  **and added an extra sentence in the discussion of the results, lines 675-677:**

  **"*The fine fuel moisture and duff moisture codes showed significant influences on ignition timing, while the drought code did not, possibly due to its slower response to changes in weather (hypothesis 5)*"**

- (Lines 481-482) You can abbreviate CAPE here.

  **Thank you for noticing this. We have changed it in the revised manuscript.**

- (Lines 535-537) Fix this grammatical error, there are two sentences here that need to be split. Also, these processes describe the extensive eastern Canadian fires in 2023, and the authors may want to mention that, although it's not strictly necessary.

  **We split up the sentence into two separate sentences in the revised manuscript. Also, we appreciate the comment on potentially adding a sentence on the recent 2023 fires in Canada (lines: 584-585):**

  **"*These processes may also have influenced the fire extremes across Canada in 2023.***

- (Lines 617-622) These discussions neglect to mention that the southern and eastern ecoregions have a much higher deciduous proportion than the northwestern ecoregions. Although this is

outside of the scope of the study, it's necessary to discuss that these are a factor influencing the drivers of early-season flammability, particularly in the Boreal Plains.

**We included this in our discussion for the final revised paper as this was an important factor for ignition efficiency, lines 668-673.**

*"The ecoregions with later snow disappearance timing, which showed less carry-over effect of snow disappearance timing on weather and fuel moisture to ignition, also corresponded to the more densely populated regions with larger fraction of deciduous trees (Pavlic et al., 2007; Olthof et al., 2015). The lack of carry-over effect may be due other drivers such as elevated potential for anthropogenic ignitions and the spring window during which deciduous forests are most flammable (Wotton et al., 2010; Parisien et al., 2023)."*

- (Figure 1) In panel B, you have used the Ecoregions without a split in the softwood shield, this should be changed for consistency.

  **Thank you for noticing this. We added the split in the Softwood Shield in the revised manuscript.**

- (Figure 3) The caption says that the ignitions on these plots correspond to the annual 20th percentile of ignitions, but some of the ecoRegions have ignitions extending past the 300th julian day. How is this possible? Is the caption wrong, or have you accidentally included all the ignitions in some of the plots? Please correct.

  **We had a deeper look into this. The 20$^{th}$ percentile was calculated annually. Some years had very few ignitions, which resulted in the 20$^{th}$ percentile occurring late in the fire season. This held true for Western Cordillera and Cold Deserts, hence, the large variation in the day of ignition. We also mainly observed this pattern in the southern ecoregions indicating that some of these small fires were due to anthropogenic activity.**

- (Figure 4) You are missing an "M" label in panel B. Panel D is not readable, please increase the size of the points significantly or change the points to their corresponding letter label (e.g. "A", "B").

  **We appreciate the suggestion. We increased the size of the points in Panel D of a revised figure 4.**

- (Figure 5) The grey dashed line is too light and difficult to read. All dashed lines should be darkened and plotted on top of the data. Also, I'm assuming the coloured fire data represents early-season ignitions, while the grey data represents all other ignitions. If this is true, why do some coloured lines appear mixed into the late-season data (e.g. in panel L)? Please fix or clarify.

  **Thanks for the comment. We have revised the figure accordingly. Indeed, the colored bars represent early ignitions and the grey bars represent the late ignitions. We made it clear in the revised figure caption what the different color schemes represent. Also, the 20$^{th}$ percentile ignition was set annually as it could differ given different weather conditions or snow conditions which caused the bars to mix with later ignition days. We have clarified this in the methods on lines 214 to 216.**

  *"Thus, we set the ignition timing threshold to the annual 20$^{th}$ percentile of the ignition timing distribution to account for potential interannual differences in weather and snow disappearance timing interfering with the ignition timing."*

**And in the figure caption for figure 5, lines 766-767:**

*"The 20$^{th}$ percentile day of ignition was set as threshold to discriminate between early (colored) and late season fires (gray)."*

- (Figure 6) Why are you reporting M-R2 and C-R2 separately for fire weather and for weather? Were these two models run independently? If so, please indicate it somewhere.

    **Indeed, since fire weather variables are derived from weather variables, we decided to run the models separately. We agree this was not clear from line 345-346.**

    *"As the pSEMs can consist of many different linear models, we fitted each component of the pSEM with a linear mixed-effect model.."*

    **We have included two sentences to clarify this, lines 346-348:**

    *"Therefore, the influence of fire weather and weather on ignition timing were modelled separately. We included the influence of snow disappearance timing in the model that contained weather variables predicting ignition timing."*

    **We have also clarified this in the figure caption, line 775:**

    *"The influence of fire weather and weather on ignition timing were modelled separately."*

- (Works cited) Most of your academic citations do not have a publication year, please add it.

    **We have looked at the references and could not find any references without a publication year. The perception of lack of publication years may stem from the journal's reference style, which places the publication year in the end, see for example, year in bold:**

    *"Abatzoglou, J. T. and Williams, A. P.: Impact of anthropogenic climate change on wildfire across western US forests,*

    *Proc. Natl. Acad. Sci. U. S. A., 113, 11770–11775, https://doi.org/10.1073/pnas.1607171113, **2016**".*

**Works cited:**

Guindon, L., Gauthier, S., Manka, F., Parisien, M.A., Whitman, E., Bernier, P., Beaudoin, A., Villemaire, P. and Skakun, R., 2021. Trends in wildfire burn severity across Canada, 1985 to 2015. *Canadian Journal of Forest Research*, *51*(9), pp.1230-1244.

Hanes, C.C., Wang, X., Jain, P., Parisien, M.A., Little, J.M. and Flannigan, M.D., 2019. Fire-regime changes in Canada over the last half century. *Canadian Journal of Forest Research*, *49*(3), pp.256-269.

Hu, L. T., & Bentler, P. M. (1999). Cutoff criteria for fit indexes in covariance structure analysis: Conventional criteria versus new alternatives. *Structural equation modeling: a multidisciplinary journal*, *6*(1), 1-55.

Pickell, P.D., Coops, N.C., Ferster, C.J., Bater, C.W., Blouin, K.D., Flannigan, M.D. and Zhang, J., 2017. An early warning system to forecast the close of the spring burning window from satellite-observed greenness. *Scientific Reports*, *7*(1), p.14190.

Verbyla, D., 2017. ABoVE: Last Day of Spring Snow, Alaska, USA, and Yukon Territory, Canada, 2000-2016. *ORNL DAAC*.